# Calmodulin dissociates the STIM1-Orai1 complex and STIM1 oligomers

Xin Li[1], Guangyan Wu[1], Yin Yang[2], Shijuan Fu[1], Xiaofen Liu[1], Huimin Kang[1],
Xue Yang[1], Xun-Cheng Su[2,3] & Yuequan Shen [1,3]

Store-operated calcium entry (SOCE) is a major pathway for calcium ions influx into cells and has a critical role in various cell functions. Here we demonstrate that calcium-bound cal-modulin ($Ca^{2+}$-CaM) binds to the core region of activated STIM1. This interaction facilitates slow $Ca^{2+}$-dependent inactivation after Orai1 channel activation by wild-type STIM1 or a constitutively active STIM1 mutant. We define the CaM-binding site in STIM1, which is adjacent to the STIM1–Orai1 coupling region. The binding of $Ca^{2+}$-CaM to activated STIM1 disrupts the STIM1–Orai1 complex and also disassembles STIM1 oligomer. Based on these results we propose a model for the calcium-bound CaM-regulated deactivation of SOCE.

[1] State Key Laboratory of Medicinal Chemical Biology and College of Life Sciences, Nankai University, 94 Weijin Road, Tianjin 300071, China. [2] State Key Laboratory of Elemento-Organic Chemistry and College of Chemistry, Nankai University, 94 Weijin Road, Tianjin 300071, China. [3] Synergetic Innovation Center of Chemical Science and Engineering, 94 Weijin Road, Tianjin 300071, China. Xin Li and Guangyan Wu contributed equally to this work. Correspondence and requests for materials should be addressed to X.Y. (email: yangxue@nankai.edu.cn) or to X.-C.S. (email: xunchengsu@nankai.edu.cn) or to Y.S. (email: yshen@nankai.edu.cn)

As an intracellular second messenger, $Ca^{2+}$ has a pivotal role in a broad range of biological processes[1]. Store-operated calcium entry (SOCE) is the major extracellular $Ca^{2+}$ influx process in excitable and, particularly, in non-excitable cells[2, 3]. Previous studies have identified the molecular constituents of SOCE: an endoplasmic reticulum (ER) $Ca^{2+}$ sensor, the stromal interaction molecule (STIM)[4–6] and a plasma membrane (PM) pore-forming subunit, Orai[7–9]. By sensing $Ca^{2+}$ depletion within the ER, STIM undergoes oligomerization and translocates to ER–PM junctions, where it activates highly $Ca^{2+}$-selective Orai channels[10]. Influx of $Ca^{2+}$ through Orai channels provides precise $Ca^{2+}$ signals that are essential for regulating long-term responses such as gene expression and cell growth[11].

$Ca^{2+}$ entry through Orai channels modulates their activity through a process called $Ca^{2+}$-dependent inactivation (CDI), which is an important negative feedback mechanism that maintains intracellular $Ca^{2+}$ homeostasis. CDI consists of fast CDI (FCDI) and slow CDI (SCDI), which have distinct kinetics and sites of action[2, 12]. FCDI occurs within tens of milliseconds after channel activation and is controlled by $Ca^{2+}$ binding to a site located several nanometers from the pore[13, 14]. The FCDI process was regulated by various factors, including the STIM1-to-Orai1 expression ratio[15], the intracellular loop of Orai1 and the conserved negatively charged cluster in the STIM1 COOH terminus[16–18]. SCDI occurs within tens of seconds of channel activation and is driven by a global rather than local $Ca^{2+}$ rise[14, 19, 20]. The SOCE-associated regulatory factor (SARAF) has been reported to facilitate SCDI of SOCE[21] through interaction with the STIM1 C-terminal inhibitory domain (448–530)[22]. Interestingly, SARAF was not able to regulate SCDI after Orai1 channel activation by the constitutively active STIM1 mutant D76A[22]. Moreover, SARAF-dependent SCDI appears to be regulated by multiple factors, such as Caveolin, E-Syt1, Septin4 and $PI(4,5)P_2$[23].

Calmodulin (CaM) consists of an N-lobe and a C-lobe, each of which contains two $Ca^{2+}$-binding sites. CaM adopts a closed conformation in the $Ca^{2+}$-free state (apo-CaM) and an open conformation in the $Ca^{2+}$-bound state (holo-CaM)[24]. Both states are able to bind various target proteins and consequently transduce different $Ca^{2+}$ signals[25]. Ion channels are major targets of CaM regulation. For example, the CDI of voltage-gated $Ca^{2+}$ channels regulated by CaM is well established[26]. Apo-CaM is pre-associated with the IQ domain, whereas $Ca^{2+}$ influx induces subtle conformational rearrangements between CaM and the $Ca^{2+}$-inactivation region, which facilitates CDI. For Orai channels, one previous study has shown that CaM might be the $Ca^{2+}$ sensor for FCDI[27, 28] but the results of a more recent report have contradicted this argument[29].

After more than 30 years of intensive studies, the activation mechanism of SOCE has been well documented[10, 11, 30, 31]. However, the deactivation mechanism is largely unknown. In this study, we demonstrated that CaM associates with the core region of STIM1 in a $Ca^{2+}$-dependent manner. Consequently, calcium-bound CaM disrupts the STIM1–Orai1 complex and disassembles STIM1 oligomers, thereby inducing deactivation of the SOCE.

## Results

**$Ca^{2+}$-CaM interacts with STIM1.** Human STIM1 is a type I membrane protein that contains 685 amino acids. Its N-terminal region is located inside the ER and contains an EF-SAM domain, which senses $Ca^{2+}$[11]. Its C-terminal region is located in the cytosolic portion of the protein and contains the core domain SOAR/CAD/OASF/Ccb9, which associates with the Orai1 channel[32–35] (for convenience, the term SOAR is used hereafter).

To investigate whether CaM is able to interact with the cytosolic C-terminal regions of STIM1 molecules, we designed various human STIM1 constructs and transiently transfected HEK293T cells. CaM-Sepharose beads were used for a pull-down experiment. We found that two constructs (SOAR and STIM1-c2) interacted with CaM in a $Ca^{2+}$-dependent manner. Neither STIM1-c1, which perturbed SOAR formation nor STIM1-c3, STIM1-c4 and STIM1-c5, which contain residues after SOAR, associated with CaM (Fig. 1a), thus suggesting that both the correct SOAR conformation and exposed SOAR are necessary for the association of the C-terminal region of STIM1 with calcium-bound CaM ($Ca^{2+}$-CaM).

To measure the binding affinity of SOAR for CaM, recombinant SOAR and CaM proteins were purified, and isothermal titration calorimetry (ITC) experiments were carried out. Our results showed that in the presence of $Ca^{2+}$, CaM interacted with SOAR with high affinity ($K_d = 0.23 \mu M$). In the absence of $Ca^{2+}$, CaM did not bind to SOAR. As a control, we also used CaM mutants (CaM-4EF) in which glutamate was mutated to glutamine to diminish $Ca^{2+}$ loading at each $Ca^{2+}$-binding site[36]. The titration of CaM-4EF into SOAR in the presence of $Ca^{2+}$ showed no detectable binding signal (Fig. 1b). These results indicated that SOAR directly and tightly associates with $Ca^{2+}$-CaM in vitro.

We then performed co-immunoprecipitation (co-IP) and fluorescence resonance energy transfer (FRET) experiments to verify that the interaction between SOAR and CaM occurs in cells. When coexpressed with myc-SOAR, CaM-YFP was able to pull down SOAR in the presence of $Ca^{2+}$ but not in its absence (Fig. 1c, d). Furthermore, when coexpressed with myc-CaM, the ER-localized STIM1 construct (1–444), which contained the entire SOAR, pulled down overexpressed CaM (Fig. 1e) and, more importantly, endogenous CaM (Fig. 1f) after cells were treated with thapsigargin. When co-expressed with Orai1, $Ca^{2+}$ influx induced significant apparent FRET efficiency (Eapp) increases between SOAR-YFP and CaM-CFP but not between SOAR-YFP and the mutant CaM-4EF-CFP (Fig. 1g). These results indicated that $Ca^{2+}$ is essential for the association of SOAR with CaM in vivo. Together, our results indicated that SOAR interacts with $Ca^{2+}$-CaM both in vitro and in vivo.

**$Ca^{2+}$-CaM facilitates the SCDI of Orai1.** To further understand the physiological significance of the $Ca^{2+}$-dependent interaction between CaM and SOAR, we used electrophysiology to determine whether CaM participates in the regulation of Orai1 SCDI. Our results showed that including purified CaM in the pipette solution increased the extent of the SCDI of Orai1 current that was induced by SOAR and the extent of inactivation in a CaM concentration-dependent manner (Fig. 2a and Supplementary Fig. 1a). Including purified CaM-4EF mutant in the pipette solution had no effect on the Orai1 current (Fig. 2b and Supplementary Fig. 1b). In the control experiment, 10 mM EGTA abolished the CaM-regulated SCDI of Orai1 current (Supplementary Fig. 2). Next, we tested whether CaM regulated the SCDI of Orai1 current that was induced by full-length STIM1. We found that CaM but not the CaM-4EF mutant facilitated the Orai1 SCDI (Fig. 2c, d and Supplementary Fig. 1c, d). We further tested whether CaM regulated the SCDI of Orai1 current that was induced by the constitutively active STIM1 mutant D76A. Our results showed that CaM, but not a CaM-4EF mutant, produced substantial increases in the SCDI of the Orai1 current (Fig. 2e and Supplementary Fig. 1e). As CaM is notoriously difficult to knock down in cells, we instead used CaM inhibitors. W-7, a canonical chemical CaM inhibitor[37], was included in the pipette solution. As we expected, W-7 inhibited the SCDI of Orai1 current that

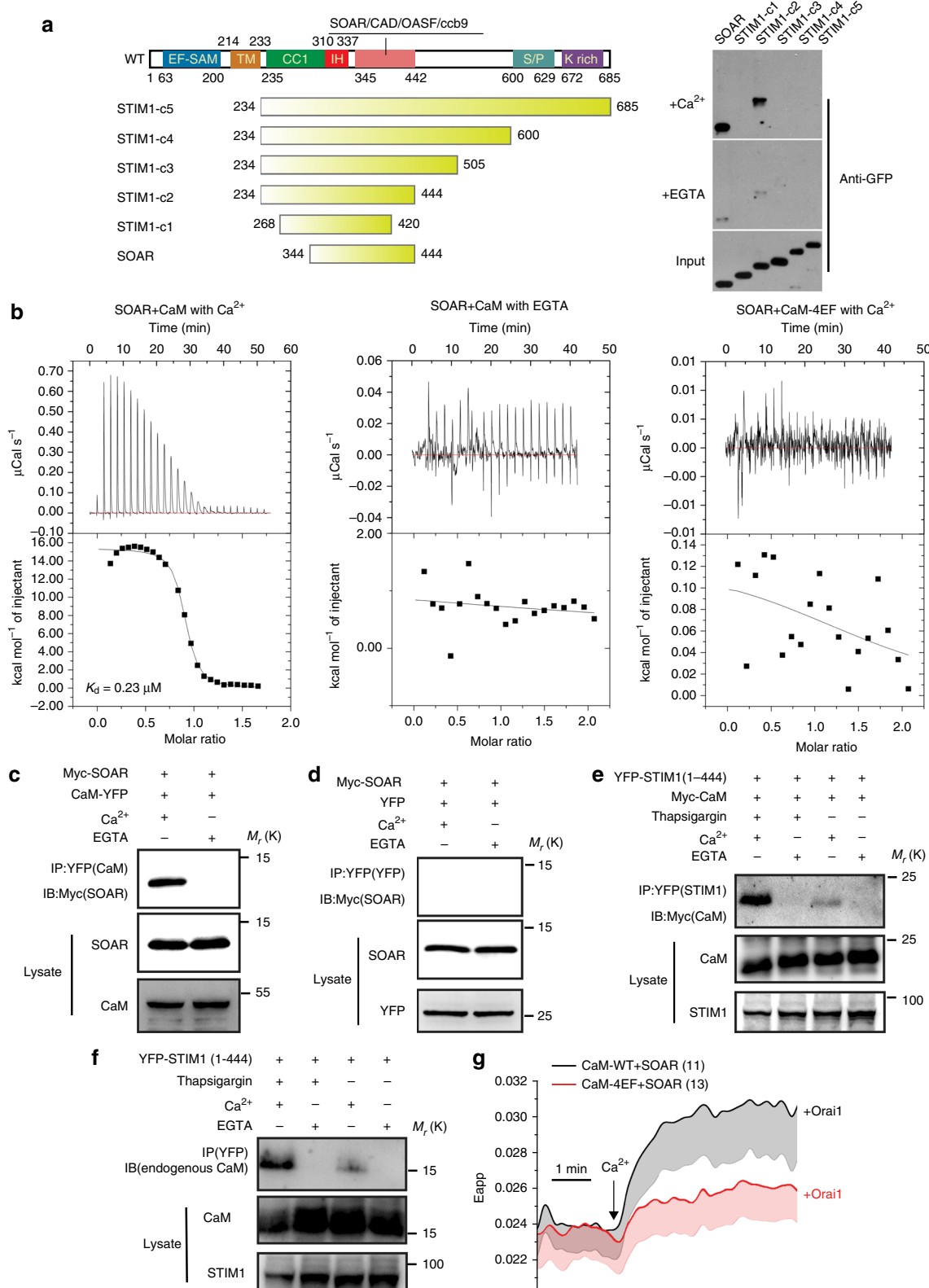

**Fig. 1** Ca$^{2+}$-CaM interacts with SOAR. **a** CaM-Sepharose pull-down of various STIM1 fragments in the presence of Ca$^{2+}$ or EGTA. **b** The dissociation constant between SOAR and CaM WT or CaM-4EF mutant in the presence of Ca$^{2+}$ or EGTA, determined by ITC. **c**, **d** Western blot analysis of co-immunoprecipitated CaM-YFP or YFP control with Myc-SOAR. Orai1 is not overexpressed in the cell. **e**, **f** Western blot analysis of co-immunoprecipitated YFP-STIM1(1–444) with overexpressed Myc-CaM or endogenous CaM. Orai1 is not overexpressed in the cell. **g** FRET between CaM-CFP or CaM-4EF-CFP (donor) and SOAR-YFP (acceptor) that are co-expressed with Myc-Orai1 in HEK293T cells. Ca$^{2+}$ was added to induce Ca$^{2+}$ influx. The number of analyzed cells is indicated. Error bars denote SEM

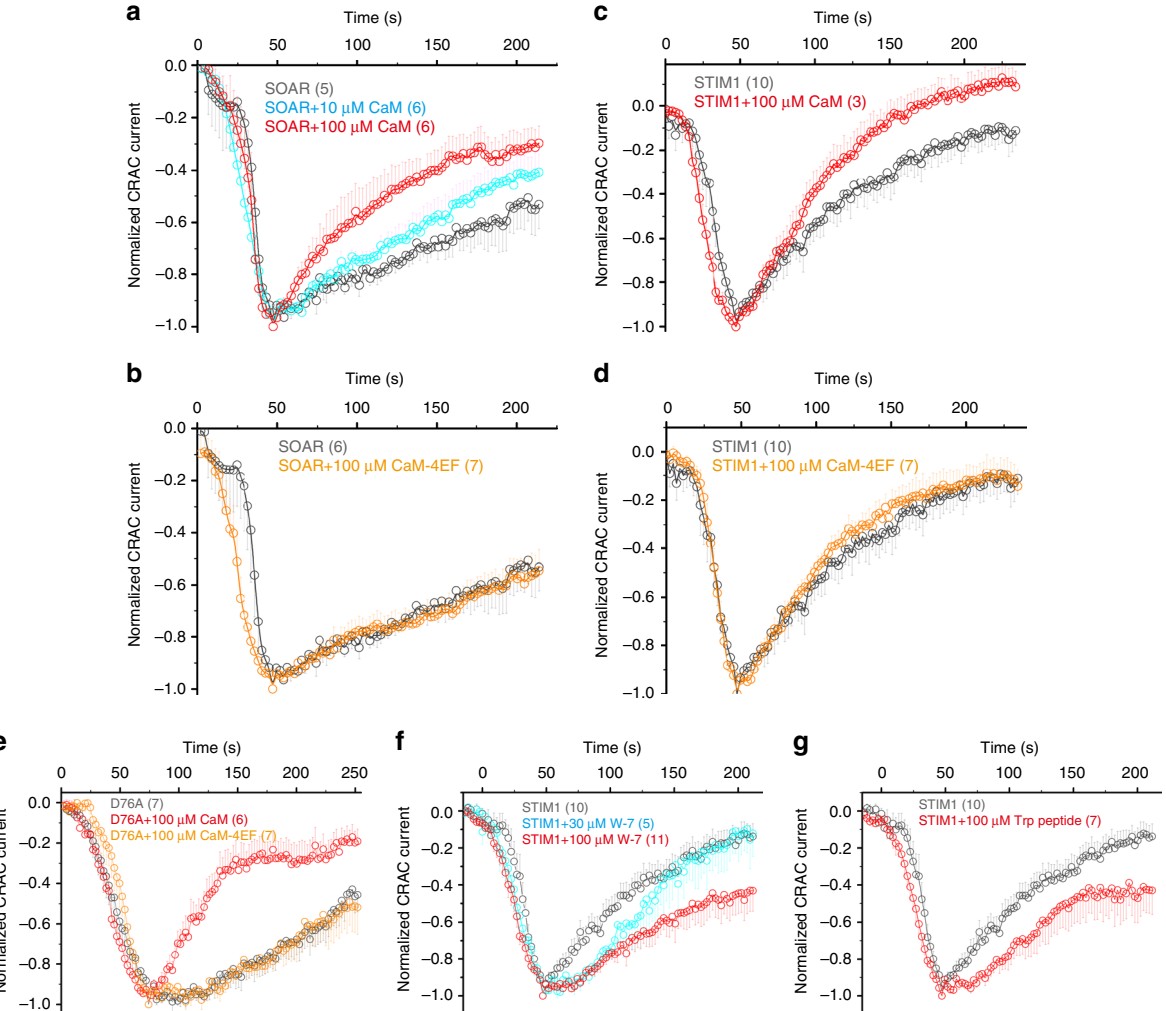

**Fig. 2** Ca$^{2+}$-CaM facilitates the SCDI of the Orai1 channel. Normalized averaged whole cell current at −100 mV measured in HEK293 cells expressing Orai1 and SOAR **a**, **b** or expressing Orai1 and STIM1 **c**, **d**. CaM and CaM-4EF mutants were introduced to the pipette solution at the indicated concentration. **e** Normalized averaged whole cell current at −100 mV measured in HEK293 cells that express Orai1 and the STIM1-D76A mutant alone, with CaM WT or with the CaM-4EF mutant at the indicated concentration. **f**, **g** Normalized averaged whole cell current at −100 mV measured in HEK293 cells expressing Orai1 and STIM1 WT with inhibitors W-7 and the TRP peptide. W-7 and the TRP peptide were introduced into the pipette solution at the indicated concentration. For all current recordings, 2 mM extracellular Ca$^{2+}$ was slowly diffused to the cell surface using a perfusion system. The number in the parenthesis indicated the number of cells that were analyzed. Error bars denote SEM

was induced by the STIM1 wild type (WT; Fig. 2f and Supplementary Fig. 1f). We then used another type of CaM inhibitor to further verify this result. The Trp peptide (RRKWQKTGHAV-RAIGRL), a CaM peptide inhibitor[38], was synthesized and included in the pipette solution. The Trp peptide significantly inhibited the SCDI of Orai1 current that was induced by the STIM1 WT (Fig. 2g and Supplementary Fig. 1g). In summary, these data suggested that CaM promotes the SCDI of the Orai1 channel.

**L390 and F391 are important in Ca$^{2+}$-CaM binding**. To address the molecular mechanism of CaM regulation of Orai1 SCDI, we sought to pinpoint the key residues that are responsible for the interaction between SOAR and Ca$^{2+}$-CaM by using high-resolution nuclear magnetic resonance (NMR) spectroscopy. Multiple human STIM1 constructs were screened, and one (STIM1 363–416) produced well-resolved peaks in the NMR $^1$H-$^{15}$N HSQC spectrum. Each residue was assigned unambiguously (Supplementary Fig. 3a). Titration of Ca$^{2+}$-CaM into the

solution of $^{15}$N-STIM1 significantly attenuated the $^{15}$N-HSQC cross-peaks for residues 382–398, thus indicating that these residues may be involved in Ca$^{2+}$-CaM binding (Fig. 3a and Supplementary Fig. 3b). This region contains several basic residues and several hydrophobic residues, which have been reported to regulate STIM1–Orai1 binding and Orai1 channel activation[30, 39, 40]. Within this region, residues L390 and F391 are good candidates to be further verified using site-directed mutagenesis, because Ca$^{2+}$-CaM typically exposes the hydrophobic pocket for target binding[24]. The $^{15}$N-HSQC experiment indicated that the interaction of Ca$^{2+}$-CaM with the $^{15}$N-labeled WT STIM1 (363–416) construct occurred with a slow-to-intermediate exchange rate, and that many cross-peaks were broadened with the addition of Ca$^{2+}$-CaM (Fig. 3b and Supplementary Fig. 3b), thus indicating that these two proteins bind each other strongly ($K_d < 10 \mu M$)[41]. In contrast, titrating Ca$^{2+}$-CaM into an $^{15}$N-labeled STIM1 (363–416) mutant, L390S-F391S, resulted in a faster exchange rate (Fig. 3b), thus indicating that mutation of L390 and F391 significantly reduced the binding affinity of the proteins. Using the cross-peak shifts of three residues (A369, G379

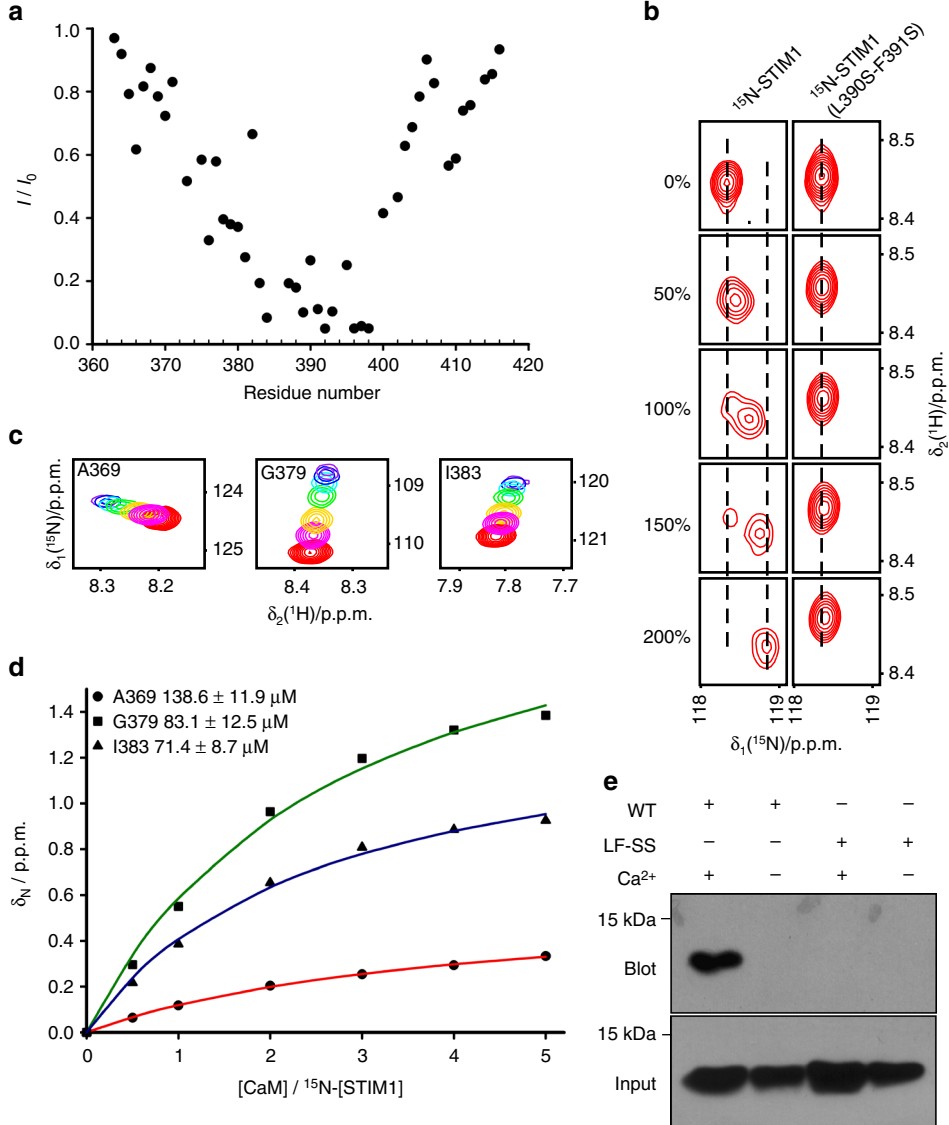

**Fig. 3** Binding site of $Ca^{2+}$-CaM to SOAR. **a** Plot of cross-peak intensity ratio, $I/I_0$, as a function of amino acid sequence, where $I$ and $I_0$ are cross-peak intensities in the $^{15}$N-HSQC spectra that were recorded for 0.1 mM $^{15}$N-STIM1 (363–416) in the presence and absence of 0.05 mM CaM, respectively. **b** Interaction comparison of $^{15}$N-STIM1 (363–416) WT and the $^{15}$N-STIM1 (363–416) L390S-F391S mutant with $Ca^{2+}$-CaM. The cross-peak of residue S400 in the $^{15}$N-HSQC spectra was monitored with respect to the [CaM]/[STIM1 (363–416) WT or mutant] molar ratio (0, 0.5, 1.0, 1.5, 2.0 from top to bottom). **c** Chemical shift changes in residues A369, G379 and I383 in the $^{15}$N-HSQC spectra recorded for 0.1 mM $^{15}$N-STIM1 (363–416) L390S-F391S mutant, with CaM concentrations ranging from 0 to 0.5 mM. **d** Dissociation constant, $K_d$, of the STIM1 (363–416) L390S-F391S mutant and CaM calculated from the chemical shift changes of backbone amide nitrogen atoms with respect to the [CaM]/[STIM1 (363–416) L390S-F391S mutant] molar ratio. **e** CaM-Sepharose pull-down experiment of SOAR WT and mutant (L390S-F391S)

and I383) as a reference, we determined the binding affinity of the STIM1 (363–416) mutant L390S-F391S with $Ca^{2+}$-CaM to be $97 \pm 10$ μM, at least 10-fold lower than that of the WT (Fig. 3c, d). It is noted that the dissociation constants that were determined by the chemical shift changes vary among different residues, perhaps reflecting the variation in local conformational changes in the association of STIM1 (363–416) with $Ca^{2+}$-CaM.

To delineate the structure of STIM1 (363–416) interacting with $Ca^{2+}$-CaM, we then performed the titration of $^{15}$N-labeled CaM-$Ca^{2+}$ with STIM1 (363–416) and the STIM1 (363–416) L390S-F391S mutant (Supplementary Fig. 4a, b). Using on the chemical shift perturbations of STIM1 and $Ca^{2+}$-CaM, the model of STIM1 (363–416) docking with $Ca^{2+}$-CaM (Supplementary Fig. 5) was generated by the HADDOCK program[42], starting with structures of $Ca^{2+}$-CaM (PDB code: 1EXR)[43] and SOAR

(PDB code: 3TEQ)[30]. Residues L390 and F391 were buried deeply inside the C-terminal lobe of $Ca^{2+}$-CaM (Supplementary Fig. 5).

We further tested the interaction between the intact SOAR domain and $Ca^{2+}$-CaM by using CaM-Sepharose bead pull-down and co-IP experiments. Our results showed that the SOAR mutant L390S-F391S (LF-SS) eliminated the interaction of SOAR with $Ca^{2+}$-CaM (Fig. 3e and Supplementary Fig. 6a), which also occurred in the STIM1 (1–444) mutant L390S-F391S (Supplementary Fig. 6b). In summary, residues L390 and F391 are necessary sites for SOAR to bind to $Ca^{2+}$-CaM.

We next asked whether the reduced affinity of the L390S-F391S mutant (SOAR–LF-SS) for $Ca^{2+}$-CaM was due to an effect on SOAR dimer formation or the α-helix structural stability that is critical for SOAR function[30, 44, 45]. Residues L390 and F391 are located on the top of the SOAR V-shaped structure

(Supplementary Fig. 7a), far from the SOAR dimerization interface[30]. A size-exclusion chromatography (SEC)-coupled multiangle light scattering experiment showed that the molecular weights of WT SOAR and LF-SS mutant proteins were similar and corresponded to the size of a dimer (Supplementary Fig. 7b, c). The co-transfection of C-terminal cyan fluorescent protein (CFP)- and yellow fluorescent protein (YFP)-labeled WT or mutant SOAR yielded comparable FRET Eapp values between WT and WT proteins, WT and mutant proteins, and mutant and mutant proteins (Supplementary Fig. 7d). These data indicated that residues L390 and F391 are dispensable in SOAR dimerization.

To test whether the SOAR–LF-SS mutant affected SOAR's α-helix structural stability, circular dichroism was used. Our results showed that WT and mutant proteins are primarily α-helical, and there were no noticeable differences between the helicities (Supplementary Fig. 7e), which is consistent with the result that $^{15}$N-STIM1 and $^{15}$N-STIM1-L390S-F391S produced similar $^{15}$N-HSQC spectra (Supplementary Fig. 8). The thermal melting curves of the two variants also showed a similar Tm value (Supplementary Fig. 7f). These data indicated that the L390S-F391S mutant did not disturb the SOAR α-helix structural stability.

**Ca$^{2+}$-CaM disrupts SOAR–Orai1 interaction**. Residues L390 and F391 are located adjacent to the polybasic patch (residues 382~387), which is the site where SOAR (and STIM1) binds to Orai1[30, 40]. Therefore, the interference of residues L390 and F391 might affect SOAR–Orai1 binding. To test this hypothesis, we co-transfected Orai1-CFP and SOAR-YFP (WT or mutant) into HEK293T cells and measured the FRET Eapp values. We observed robust FRET signals between Orai1-CFP and the WT SOAR-YFP (SOAR-WT) at the PM. In contrast, the FRET signals between Orai1-CFP and the SOAR-YFP mutant (SOAR–LF-SS) were negligible (Fig. 4a, b). Moreover, live-cell confocal microscopy showed that SOAR-WT co-localized with Orai1 on the PM, whereas the mutant SOAR–LF-SS did not (Fig. 4c), thus indicating that the mutation of residues L390 and F391 abrogated the interaction of SOAR with Orai1. We further verified this reduced interaction by performing a co-IP experiment. When coexpressed with myc-Orai1, SOAR-WT (but not SOAR–LF-SS) was able to pull down myc-Orai1 (Fig. 4d). To test whether mutating residues L390 and F391 affected Orai1 channel activation, we measured intracellular Ca$^{2+}$ influx in HEK293T cells that expressed both SOAR-WT or the SOAR–LF-SS mutant and Orai1. SOAR-WT showed constitutive Ca$^{2+}$ influx, whereas the mutant SOAR–LF-SS did not induce Ca$^{2+}$ influx (Fig. 4e). These data suggested that residues L390 and F391 are important in SOAR–Orai1 binding.

Considering that residues L390 and F391 are also critical for Ca$^{2+}$-CaM binding to SOAR, we tested whether Ca$^{2+}$-CaM could directly disrupt the SOAR–Orai1 complex in vitro. GFP-Orai1 and myc-SOAR were co-transfected into HEK293T cells. The SOAR–Orai1 complex was purified by GFP-Trap and washed with purified CaM in the absence or presence of Ca$^{2+}$. Our results showed that CaM-containing solution largely disrupted the SOAR–Orai1 complex in a Ca$^{2+}$-dependent manner, compared with CaM-free solution (Fig. 4f).

We then investigated whether Ca$^{2+}$-CaM affected the binding of SOAR to Orai1 by measuring the FRET Eapp value at the PM. Orai1-CFP, YFP-SOAR and CaM or CaM-4EF mutants were co-transfected into HEK293T cells. FRET Eapp values at the PM were measured after extracellular Ca$^{2+}$ was added. Compared with the control, the overexpression of WT CaM clearly decreased the FRET Eapp value, whereas the overexpression of the CaM-4EF mutant resulted in a similar FRET Eapp value (Fig. 4g). To inhibit endogenous CaM, Orai1-CFP and YFP-

SOAR were co-transfected into HEK293T cells, and cells were treated with the inhibitor W-7 or a dimethyl sulfoxide (DMSO) control. After 5 min of extracellular Ca$^{2+}$ influx, the FRET Eapp value at the PM was significantly higher in W-7-treated cells than in DMSO-treated cells (Fig. 4h). These data together suggested that Ca$^{2+}$-CaM dissociates the SOAR–Orai1 complex.

**Ca$^{2+}$-CaM disrupts the STIM1–Orai1 interaction**. SOAR is the minimum activation domain of STIM1 that can bypass ER Ca$^{2+}$-depletion and bind and activate Orai1[33]. Residues L390 and F391 are important in the SOAR–Orai1 interaction. We asked whether residues L390 and F391 are also important in the interaction of full-length STIM1 with Orai1. Live-cell confocal microscopy was used to monitor distribution changes of full-length STIM1-GFP (WT and mutant) between the resting and activation states when it was co-expressed with mCherry-Orai1. In the resting state, both the STIM1-WT and STIM1 mutant (STIM1–LF-SS) were distributed across the ER, whereas Orai1 was uniformly distributed on the PM (Fig. 5a). After thapsigargin stimulation, STIM1-WT translocated and co-localized with Orai1. In contrast, the mutant STIM1–LF-SS did not co-localize with Orai1 after thapsigargin stimulation (Fig. 5a). We further transiently transfected STIM1-YFP (WT or mutant) into the HEK-Orai1-CFP-stable cells and then measured FRET signals. Ca$^{2+}$ depletion from the ER induced an approximately twofold increase of FRET Eapp between Orai1 and STIM1-WT, whereas no increase was observed between Orai1 and STIM1–LF-SS (Fig. 5c). Figure 5b shows the FRET Eapp change for the same cell before and after thapsigargin stimulation. We also verified the interaction by using a co-IP experiment. When co-expressed with myc-Orai1, STIM1-WT but not STIM1–LF-SS associated with myc-Orai1 after thapsigargin treatment (Fig. 5e). In HEK293T-cells, extracellular Ca$^{2+}$ influx was abolished with the co-expression of Orai1 and the mutant STIM1–LF-SS (Fig. 5d). These data showed that residues L390 and F391 are also necessary for STIM1–Orai1 binding and Orai1 activation.

We then tested whether Ca$^{2+}$-CaM could disrupt the STIM1–Orai1 interaction. GFP-Orai1 and STIM1-myc were co-transfected into HEK293T cells. The cells were treated with ionomycin, and the STIM1–Orai1 complex was then purified by GFP-Trap. We found that CaM was able to wash STIM1 off the STIM1–Orai1 complex in a Ca$^{2+}$-dependent manner (Fig. 5f). To further determine whether Ca$^{2+}$-CaM affects the interaction of STIM1 with Orai1, Orai1-CFP, STIM1-YFP and WT CaM or the CaM-4EF mutant were co-transfected into HEK293T cells, and the FRET Eapp values were measured after 5 min of ionomycin treatment. As expected, the overexpression of WT CaM but not the CaM-4EF mutant largely decreased the FRET values compared with those of the control (Fig. 5g). When endogenous CaM was inhibited by the W-7 inhibitor, the FRET Eapp value increased significantly compared with that of the DMSO control (Fig. 5h). Together, these data indicated that Ca$^{2+}$-CaM dissociates the STIM1–Orai1 complex.

**Ca$^{2+}$-CaM disassembles the STIM1 oligomer**. Puncta formation at ER-PM junctions is a hallmark and key step for STIM1 activation[46–48]. We therefore determined whether mutation of residues L390 and F391 would affect the formation of STIM1 puncta. We monitored puncta formation by using live-cell confocal microscopy in HEK293T cells that expressed STIM1-GFP (WT or mutant) alone or with mCherry-Orai1. In cells expressing STIM1 alone, the WT and mutant were distributed across the ER in resting cells (Fig. 6a, upper panel). After stimulation by thapsigargin, STIM1 WT translocated and formed puncta, whereas no puncta were observed for cells with the mutant

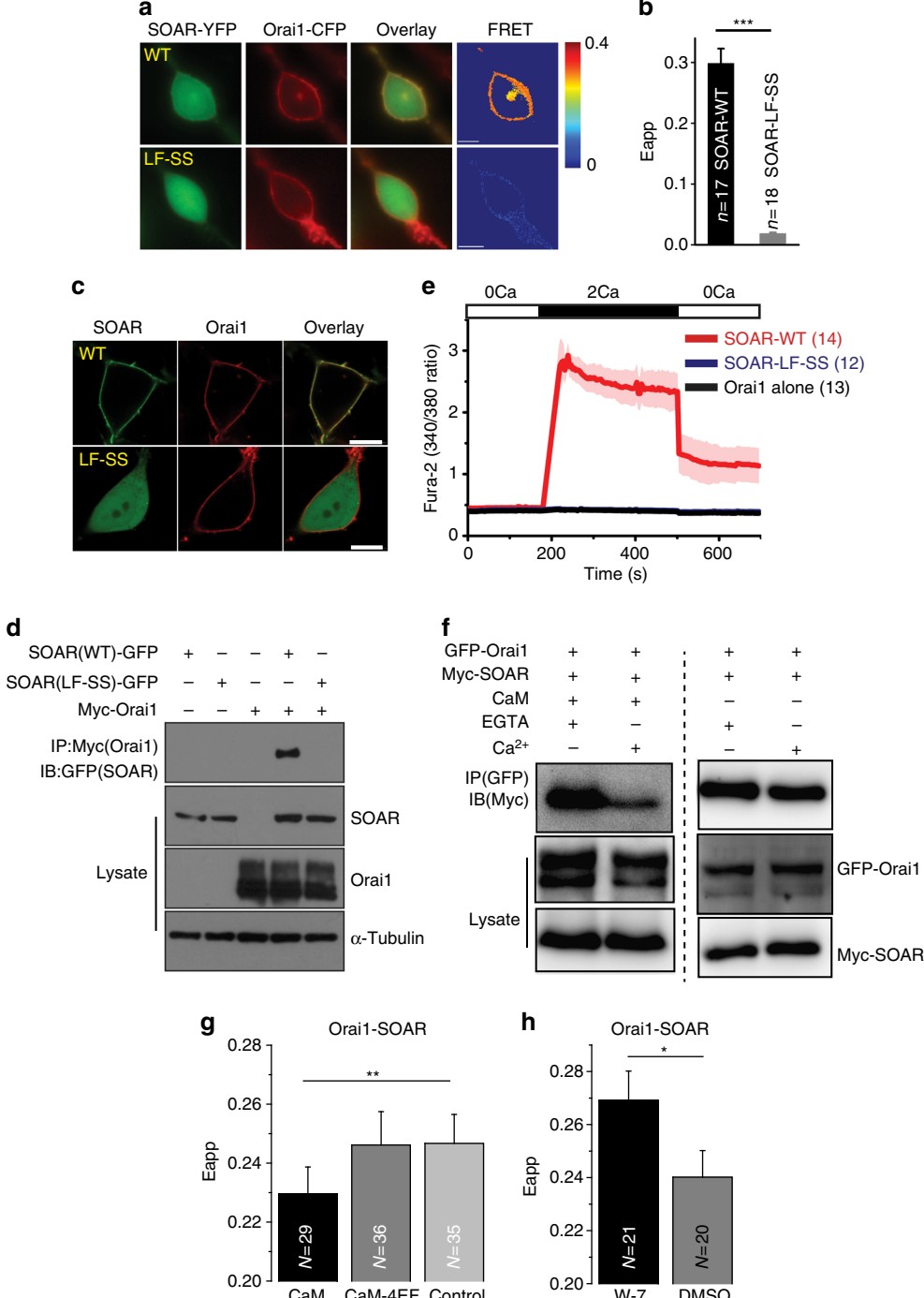

**Fig. 4** Ca$^{2+}$-CaM dissociates the SOAR–Orai1 complex. **a, b** Resting FRET was measured between Orai1-CFP and SOAR (WT or mutant)-YFP on PM of HEK Orai1-CFP stable cells. The cells were in 0 mM extracellular calcium Ringer's buffer. Scale bars are 10 μm. **c** Confocal images of live HEK293T cells transfected with SOAR-GFP (WT or mutant) and Cherry-Orai1. Scale bars are 10 μm. **d** Western blot analysis of co-immunoprecipitated SOAR-GFP (WT or mutant) with Myc-Orai1. **e** Ca$^{2+}$ influx in HEK293T cells co-expressing Cherry-Orai1 and SOAR-YFP WT or mutant. **f** Western blot analysis of competitive binding of the SOAR–Orai1 complex in the presence or absence of Ca$^{2+}$-CaM. **g** Steady-state FRET between Orai1-CFP and YFP-SOAR co-expressed with CaM or CaM-4EF mutant on PM of HEK293T cells. FRET values were measured after 2 mM extracellular Ca$^{2+}$ was added. **h** Steady-state FRET between Orai1-CFP and YFP-SOAR on the PM of HEK293T cells that were pretreated with 30 μM W-7 or DMSO control for 30 min. FRET values were measured after 2 mM extracellular Ca$^{2+}$ was added. Numbers of cells that were analyzed are indicated. *$P < 0.05$, **$P < 0.01$ (unpaired Student's $t$-test). Error bars denote SEM

STIM1–LF-SS (Fig. 6a, lower panel). In cells that co-express Orai1, after thapsigargin stimulation, the STIM1 WT formed puncta that were co-localized with Orai1 near the PM. The mutant STIM1–LF-SS neither changed distribution nor co-localized with Orai1 (Fig. 6b).

STIM1 exists as a dimer in the resting state, and Ca$^{2+}$ depletion from ER store induces further oligomerization[2, 46]. The STIM1–LF-SS mutant did not form puncta, thus suggesting that this mutant is unable to self-oligomerize after store depletion. To further examine this hypothesis, we used FRET microscopy

methods. We co-transfected STIM1-CFP and STIM1-YFP (WT or mutant) into HEK293T cells. In the resting state, the STIM1 WT and mutant STIM-LF-SS showed similar FRET Eapp values. These values were higher than the control values, thus indicating that the mutation of residues L390 and F391 did not impair STIM1 dimer formation (Fig. 6c). $Ca^{2+}$ depletion from ER stores induced a robust increase in the FRET Eapp value for the STIM1 WT, owing to STIM1 oligomerization, in agreement with results from previous reports[44, 46, 49]. In contrast, the FRET Eapp value remained similar for the mutant STIM1–LF-SS, thereby indicating that the mutation of residues L390 and F391 prevented STIM1 oligomerization (Fig. 6c).

As residues L390 and F391 are important in STIM1 oligomerization, we hypothesized that $Ca^{2+}$-CaM might have the capability to disassemble the STIM1 oligomer. STIM1-YFP and STIM1-myc were co-transfected into HEK293T cells. After ionomycin treatment, STIM1 oligomers were purified by GFP-Trap and incubated with $Ca^{2+}$-CaM. We found that $Ca^{2+}$-CaM largely reduced STIM1 oligomerization in our in vitro system (Fig. 6d). To investigate whether $Ca^{2+}$-CaM would affect STIM1 oligomerization in cells, STIM1-CFP, STIM1-YFP and WT CaM or CaM-4EF mutants were co-transfected into HEK293T cells. FRET Eapp values were measured after 5 min of ionomycin treatment. The overexpression of CaM WT resulted in decreased

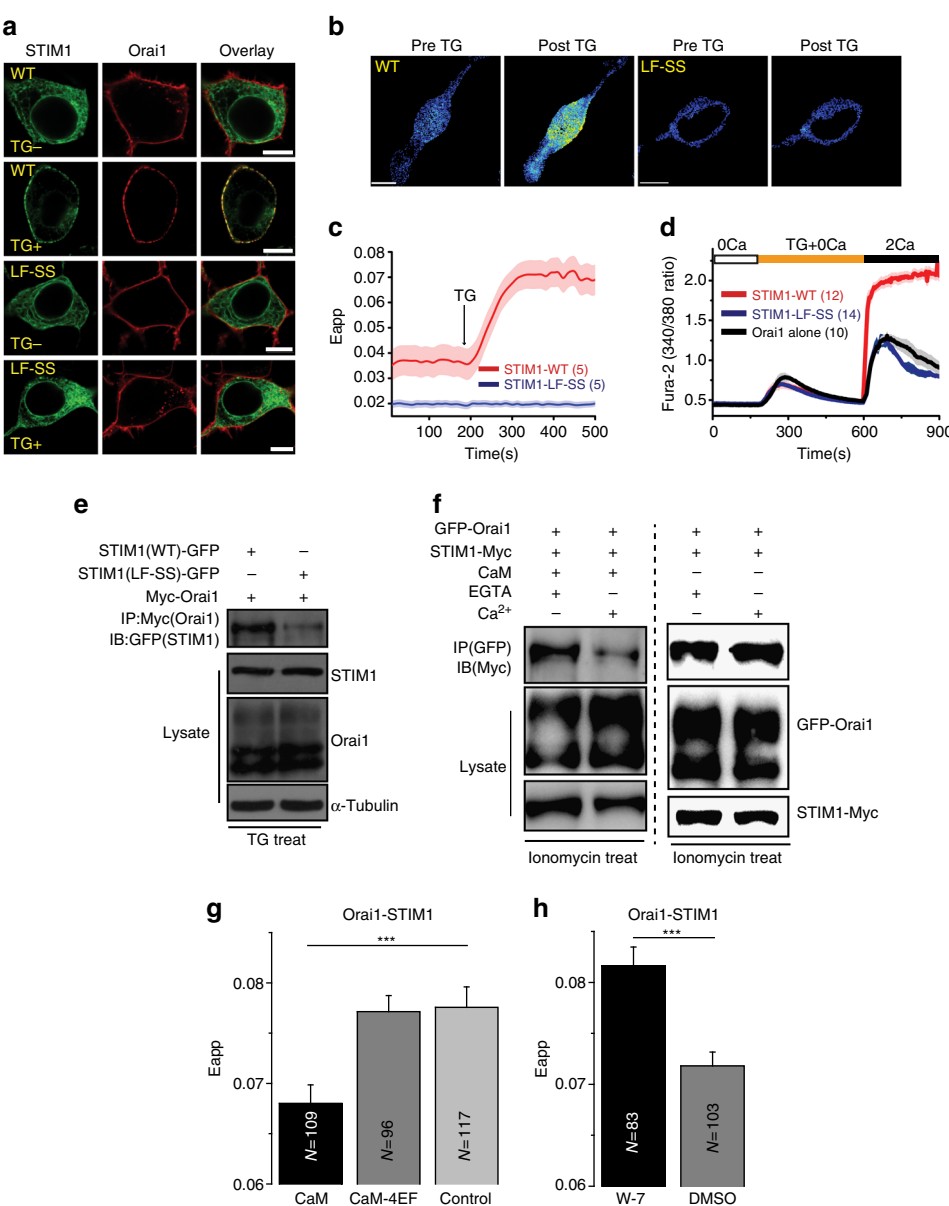

**Fig. 5** $Ca^{2+}$-CaM dissociates the STIM1–Orai1 complex. **a** Confocal images of live HEK293T cells transfected with STIM1-GFP (WT or mutant) and Cherry-Orai1 before and after 1 μM thapsigargin (TG) stimulation. Scale bars are 10 μm. **b, c** FRET was measured between Orai1-CFP and STIM1-YFP (WT or mutant). TG (1 μM) was used to induce $Ca^{2+}$ depletion. The same cell before and after TG stimulation is shown. Scale bars are 10 μm. **d** $Ca^{2+}$ influx in HEK293T cells co-expressing Cherry-Orai1 and STIM1-YFP WT or mutant. **e** Western blot analysis of co-immunoprecipitated STIM1-GFP (WT or mutant) with Myc-Orai1. TG (1 μM) was used to induce $Ca^{2+}$ depletion before lysis. **f** Western blot analysis of competitive binding of $Ca^{2+}$-CaM protein with STIM1–Orai1 complex in HEK293T cells. Ionomycin (3 μM) was used to induce $Ca^{2+}$ depletion before lysis. **g** Steady-state FRET between Orai1-CFP and STIM1-YFP that were co-expressed with CaM or CaM-4EF mutant after 3 μM ionomycin treatment in HEK293T cells. **h** Steady-state FRET between Orai1-CFP and STIM1-YFP in HEK293T cells that were pretreated with 30 μM W-7 or DMSO control for 30 min after 3 μM ionomycin treatment. Numbers of cells analyzed are indicated. ***$P < 0.001$ (unpaired Student's t-test). Error bars denote SEM

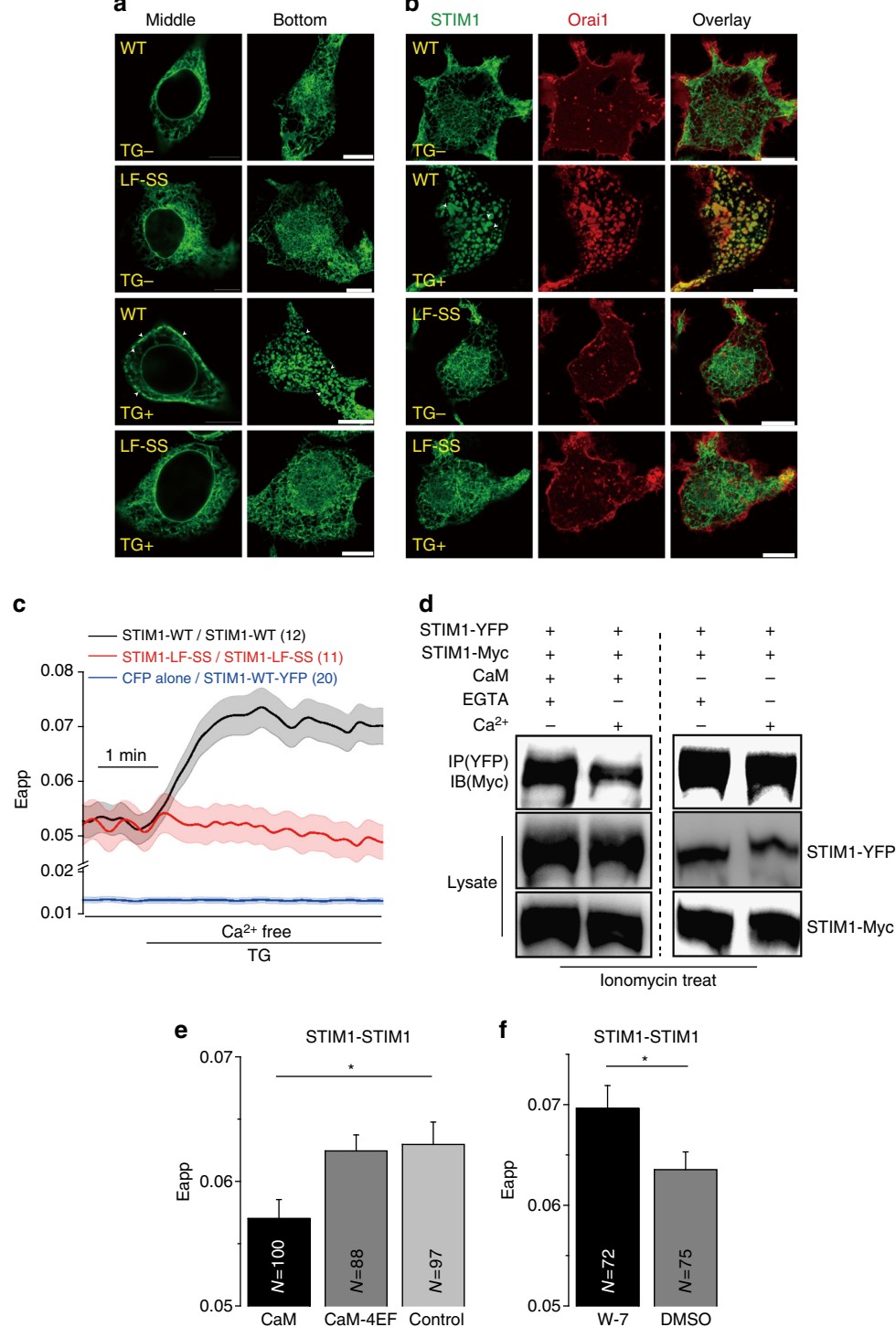

**Fig. 6** Ca²⁺-CaM disassembles the STIM1 oligomer. **a**, **b** Confocal images of live HEK293T cells that were transfected with STIM1-GFP (WT or mutant) alone or co-transfected with Cherry-Orai1 before and after 1 μM TG stimulation. Puncta are indicated by the arrow. Scale bars are 10 μm. **c** FRET was measured between WT or mutant with COOH-terminus CFP- and YFP-tagged STIM1 in HEK293T cells. TG (1 μM) was used to induce Ca²⁺ depletion. **d** Western blot analysis of Ca²⁺-CaM protein competitive binding with STIM1 intermolecular interactions in HEK293T cells. Ionomycin (3 μM) was used to induce Ca²⁺ depletion before lysis. **e** Steady-state FRET between STIM1-CFP and STIM1-YFP that were co-expressed with CaM or CaM-4EF mutant in HEK293T cells after 3 μM ionomycin treatment. **f** Steady-state FRET between STIM1-CFP and STIM1-YFP pretreated with 30 μM W-7 or DMSO control for 30 min after 3 μM ionomycin treatment in HEK293T cells. Numbers of cells analyzed are indicated. *$P < 0.05$ (unpaired Student's *t*-test). Error bars denote SEM

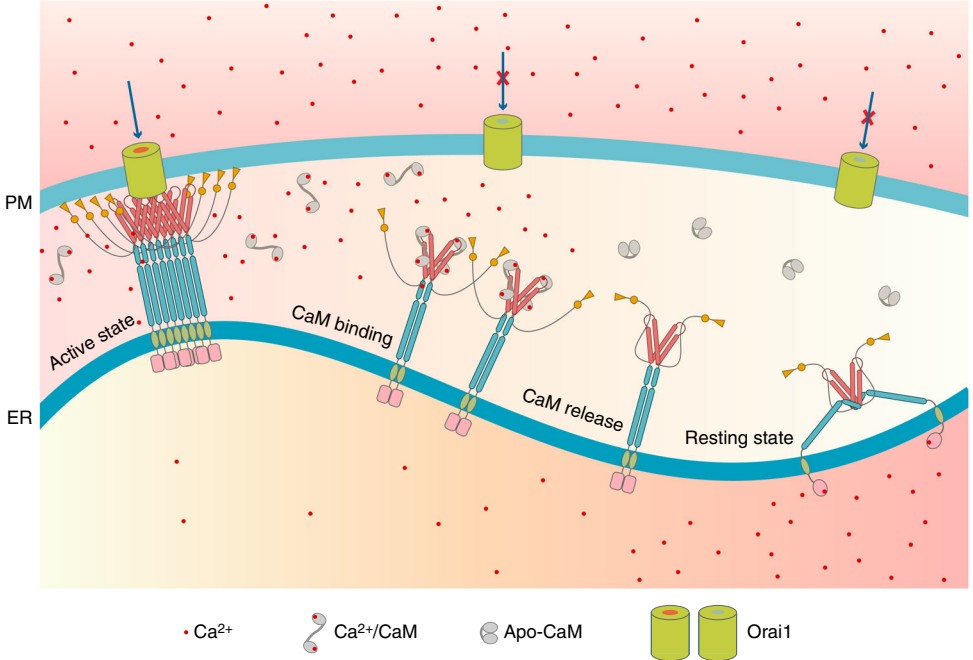

**Fig. 7** Model of STIM1–Orai1 deactivation induced by $Ca^{2+}$-CaM. Extracellular $Ca^{2+}$ influxes after the formation of the STIM1–Orai1 complex, leading to elevation of cytosolic $Ca^{2+}$. CaM undergoes a conformational change and binds to the exposed SOAR region of STIM1. The STIM1–Orai1 complex dissociates, the Orai1 channel is inactivated and the STIM1 oligomer is disassembled. As the cytosolic $Ca^{2+}$ concentration is decreased, CaM dissociates from STIM1. STIM1 returns to the resting state, owing to a filled ER $Ca^{2+}$ store and possibly other regulators

FRET values, whereas the overexpression of CaM-4EF mutant did not (Fig. 6e). The inhibition of endogenous CaM by W-7 led to FRET values that increased compared with those of the DMSO control (Fig. 6f). These data suggest that $Ca^{2+}$-CaM can disassemble the STIM1 oligomer.

## Discussion

SOCE is a major cell signaling path that has an essential role in various cell activities[3, 11, 50]. The process by which ER resident protein STIM1 senses $Ca^{2+}$ changes inside the ER and consequently regulates the PM-located $Ca^{2+}$ channel Orai1 has attracted the attention of many researchers for many decades. The molecular mechanism of STIM1 activation coupling to Orai1 has been elucidated, whereas the process from the activated STIM1–Orai1 complex to the resting state remains elusive. Our studies revealed that calcium-bound CaM associates with an activated STIM1 in which SOAR is exposed and consequently dissociates the STIM1–Orai1 complex and the STIM1 oligomer, thus reversing the process by which STIM1 activates Orai1.

CaM has a crucial role in maintaining $Ca^{2+}$ homeostasis by regulating many $Ca^{2+}$ channels and pumps, such as voltage-dependent $Ca^{2+}$ channels and PM $Ca^{2+}$-ATPase[51, 52]. Our results showed that CaM interacts with SOAR in a $Ca^{2+}$-dependent manner in vitro and in vivo. In particular, the STIM1 construct (1–444) was able to pull-down both overexpressed CaM and endogenous CaM after cells were stimulated by thapsigargin, thus providing good evidence that $Ca^{2+}$-CaM has the capability to associate with exposed SOAR in vivo. Moreover, using a FRET technique, we observed that extracellular $Ca^{2+}$ influx induced CaM access to SOAR. The control CaM-4EF mutant was not able to access SOAR. These data suggest that after extracellular $Ca^{2+}$ influx causes cytosolic $Ca^{2+}$ elevation, CaM senses and loads with $Ca^{2+}$ and then accesses and binds to SOAR in activated STIM1.

Interestingly, the SOAR region does not contain a known CaM-binding motif. We then searched for the CaM-binding site in SOAR in a CaM target database[53] and found that the region around L390-F391 represents the most likely binding site. We further located the specific binding site through NMR titration and validated this binding site through immunoprecipitation methods. So far, two regions in STIMs were found to interact with $Ca^{2+}$-CaM[54, 55]. One region is located in the STIM1 C-terminus and the other is located in the STIM2 SOAR domain. First, the very C-terminal polybasic region (667–685) in STIM1 was found through ITC and NMR methods to interact with $Ca^{2+}$-CaM. This region (667–685) did not contain any known CaM-binding site and consists of several basic and hydrophobic residues. ITC experiments in which the polybasic peptide was titrated into CaM revealed a strong binding affinity between these two molecules (1 μM)[54]. Second, through pull-down and SPR methods, the region 459–482 in SOAR of STIM2, which corresponds with residues 368–391 in SOAR of STIM1, was found to interact with $Ca^{2+}$-CaM[55]. This region is also rich in basic and hydrophobic residues and did not contain any known CaM-binding motif. Therefore, CaM seems to bind with STIM1 in an unconventional way.

SOAR is the basic unit of STIM1 that binds and activates the Orai1 channel. It has been demonstrated that the polybasic SOAR residues 384–387 are crucial for SOAR binding to Orai1. Interestingly, we identified residues L390 and F391 as necessary for SOAR binding to $Ca^{2+}$-CaM. These residues are adjacent to the Orai1-binding site of SOAR, thus suggesting that $Ca^{2+}$-CaM binding to SOAR will interfere with the SOAR–Orai1 complex or the STIM1–Orai1 complex. Furthermore, the affinity value that we measured between SOAR and $Ca^{2+}$-CaM was ~0.2 μM, ~1,000-fold higher than the affinity (~200 μM) of SOAR for the Orai1 channel, according to previous reports[45, 56]. This large difference in affinity allows $Ca^{2+}$-CaM to compete with Orai1 for association with SOAR, thus leading to the breakdown of the SOAR–Orai1 complex or the STIM1–Orai1 complex.

Puncta formation represents STIM1 aggregation and activation. Our results showed that the mutation of residues Leu390

and Phe391, which are part of the $Ca^{2+}$-CaM binding site of SOAR or STIM1, leads to the failure of puncta formation and activation. Thus, we infer that $Ca^{2+}$-CaM binding to the activated STIM1 oligomer may lead to a shift in equilibrium from an STIM1 oligomer to a less aggregated form. In our system that uses purified CaM, we indeed observed that the STIM1 oligomer became less aggregated after the addition of $Ca^{2+}$-CaM. This result was further confirmed by FRET measurements in cells.

The Orai1 channel is characteristic of both FCDI and SCDI. FCDI occurs within milliseconds, whereas SCDI occurs within minutes. Our results showed that after CaM responds to elevated cytosolic $Ca^{2+}$ levels, $Ca^{2+}$-CaM disrupts the SOAR–Orai1 complex or the STIM1–Orai1 complex, resulting in the inactivation of the Orai1 channel. This process was consistent with the SCDI of the Orai1 channel. Indeed, including purified CaM in the pipette facilitated Orai1 SCDI. More importantly, when the Orai1 channel was activated by the constitutively active STIM1 mutant D76A, CaM was also able to facilitate SCDI. When inhibitors were used to inhibit endogenous CaM, the SCDI of the Orai1 current was reduced. Thus, $Ca^{2+}$-CaM induced SCDI appears to differ from SARAF-induced SCDI. It has been reported that SARAF-induced SCDI may be mediated by the interaction of SARAF with residues 448–530 of STIM1, which are located after SOAR[22]. In addition, STIM1 and SARAF are localized in the ER and interact in the basal state with filled $Ca^{2+}$ stores[21, 23], suggesting that SARAF functions by holding STIM1 in a resting state. This statement is consistent with data that SARAF does not inactivate the Orai1 channel if the constitutively active STIM1 mutant D76A or 4E-4A is used[22]. Therefore, we speculate that $Ca^{2+}$-CaM has a dominant role in disrupting the STIM1–Orai1 complex and disaggregating the STIM1 oligomer. The inactive STIM1 conformation is presumably maintained by SARAF to avoid the spontaneous activation of STIM1. More evidence is required to test this possibility in the near future.

In summary, we propose a molecular mechanism of STIM1–Orai1 deactivation that is induced by $Ca^{2+}$-CaM (Fig. 7). After Orai1 is activated by STIM1 and forms the STIM1–Orai1 complex, extracellular $Ca^{2+}$ influx through the Orai1 channel leads to the elevation of cytosolic $Ca^{2+}$. As the cytosolic $Ca^{2+}$ increases to a certain level, CaM is charged with $Ca^{2+}$ and moves closer to the STIM1–Orai1 complex. Owing to the high affinity between $Ca^{2+}$-CaM and the exposed SOAR, $Ca^{2+}$-CaM pulls STIM1 from Orai1 and disrupts the STIM1–Orai1 complex. The Orai1 channel is then closed. Furthermore, the binding of $Ca^{2+}$-CaM to STIM1 induces a change in the STIM1 oligomer that results in a less-aggregated form. By the action of ER $Ca^{2+}$-ATPase and PM $Ca^{2+}$-ATPase, the cytosolic $Ca^{2+}$ concentration is decreased, and the ER $Ca^{2+}$ store is refilled. Decreased cytosolic $Ca^{2+}$ concentrations lead to CaM dissociation from STIM1, whereas increased ER $Ca^{2+}$ concentrations load the N-terminal STIM1 region with $Ca^{2+}$. Consequently, the N-terminal STIM1 returns to its monomeric form and presumably drives STIM1 back to the resting state. Of course, this process can be further regulated by many other factors, such as SARAF[23], the orosomucoid-like 3 protein ORMDL3[57] and Golli[58].

Our model is consistent with previously reported results. In RBL cells, SCDI is mainly regulated by cytosolic $Ca^{2+}$ concentration[14, 19], indicating that cytosolic $Ca^{2+}$ concentrations have a critical role in regulating the SCDI. $Ca^{2+}$ influx through the Orai1 channel negatively regulates channel activity by promoting dissociation of the STIM1–Orai1 complex and subsequent STIM1 deoligomerization to the resting state[46, 59]. ER $Ca^{2+}$ replenishment is not sufficient to disassemble STIM1 oligomers and requires cytosolic $Ca^{2+}$ concentration elevation[59]. It has been reported that the ER protein ORMDL3 binds to and inhibits the

SERCA pump, promoting cytosolic calcium accumulation and resulting in negatively regulated SOCE[60]. It is likely to be that increased cytosolic $Ca^{2+}$ levels activate CaM and facilitate the SCDI of the Orai1 channel. CaM has been proposed to be involved in FCDI through the interaction of CaM with the N-terminal residue W76 of the Orai1 channel[61, 62]. A recent study has argued against this model because W76 is located in the pore of the Orai1 channel and is thus unlikely to be accessed by CaM[29]. Whether CaM is involved in FCDI may require further assessment. Thus, FCDI is beyond the scope of this manuscript.

In combination with previously described STIM1–Orai1 activation processes, our results now depict the entire SOCE at the molecular level. $Ca^{2+}$ has a central role in initiating, directing and terminating the SOCE.

## Methods

**Cell culture and transfection**. HEK293T (ATCC) were cultured in Dulbecco's modified Eagle's medium (DMEM) (Sigma-Aldrich) supplemented with 10% fetal bovine serum (Hyclone). HEK stable expressing Orai1-CFP (gift from Dr Youjun Wang from Beijing Normal University) were cultured in DMEM supplemented with 10% fetal bovine serum and 100 μg ml$^{-1}$ G418. All cells were maintained in 37 °C, 5% $CO_2$ humidified incubator. All plasmids were transfected using Fugene6 (Promega) according to the manufacturer's instructions.

**Plasmids**. Full-length complementary DNA of human STIM1 was inserted into pECFP-C1, pEGFP-C1 and pEYFP-C1 vectors (Clontech) between BglII and EcoRI sites to generate STIM1-CFP, STIM1-GFP and STIM1-YFP. The sequence of SOAR was amplified by PCR and cloned into pCMV-Myc (Clontech) vector between EcoRI and BglII sites to generate the Myc-SOAR. Full-length cDNA of human Orai1 was inserted into pmCherry-C1 vector (Clontech) between BglII and EcoRI sites to generate mCherry-Orai1. Full-length cDNA of human Orai1 was inserted into pCMV-Myc vector between EcoRI and BglII sites to generate Myc-Orai1. GFP-STIM1 truncations were generated by introducing stop codon at the specific position. Full-length cDNA of rat CaM was cloned into pET-33(b) vector (Novagen) between NcoI and XhoI. mCherry-Orai1, SOAR-GFP, myc-Orai1, myc-SOAR and MBP-SOAR have been previously described[30]. Mutations were generated by using the Quick-change mutagenesis kit (Agilent). Primers used in this study were listed in Supplementary Table 1.

**Protein expression and purification**. Human SOAR WT and LF-SS mutant were expressed in *Escherichia coli*. The cells were collected and suspended in lysis buffer (20 mM MES pH 6.0, 1 mM EGTA and 300 mM NaCl) and lysed by sonication. After centrifugation at 15,000 $g$ for 1 h, the supernatant was loaded onto a MBP-Trap HP column (GE Healthcare) equilibrated with equilibration buffer (20 mM MES pH 6.0, 300 mM NaCl and 1 mM EGTA). The target protein was eluted using elution buffer (20 mM MES pH 6.0, 300 mM NaCl, 1 mM EGTA, 10 mM Maltose) and cleaved with Tobacco Etch Virus protease overnight at 4 °C. The protein was applied to a HiTrap SP column (GE Healthcare) equilibrated with 20 mM MES pH 6.0, 50 mM NaCl, 1 mM dithiothreitol (DTT) and 1 mM EGTA and eluted by increasing the NaCl concentration. Target protein was collected and further purified by using a Superdex 200 10/300 GL column (GE Healthcare). Protein concentrations were determined using a BCA assay kit (Pierce). The purification of the LF-SS mutant followed the same protocol for the SOAR WT.

CaM and CaM-4EF were expressed in *E. coli*. The cells were collected and suspended in lysis buffer (50 mM Tris-HCl pH 7.5, 1 mM EGTA and 1 mM DTT) and lysed by sonication. For the WT CaM, after centrifugation at 15,000 $g$ for 1 h, the supernatant was supplemented with NaCl and $CaCl_2$ to a final concentration of 500 mM NaCl and 5 mM $CaCl_2$. The supernatant was loaded onto a HiPrep Phenyl FF (High sub) 16/10 column (GE Healthcare) equilibrated with equilibration buffer (50 mM Tris-HCl pH 7.5, 500 mM NaCl and 5 mM $CaCl_2$). Protein was eluted using elution buffer (50 mM Tris-HCl pH 7.5, 1 mM EGTA) and further purified by using a HiLoad 26/600 Superdex 200 column (GE Healthcare). For CaM-4EF, the supernatant was applied to a HiPrep Q FF 16/10 column (GE Healthcare) equilibrated with 50 mM Tris-HCl pH 7.5, 50 mM NaCl, 1 mM DTT and 1 mM EGTA. Protein was eluted by increasing the NaCl concentration. Target protein was collected and further purified by using a HiLoad 26/600 Superdex 200 column (GE Healthcare). Protein concentrations were determined using a BCA assay kit (Pierce).

The STIM1(363–416) construct was expressed in *E. coli*. The cells were collected and suspended in lysis buffer (50 mM Tris-HCl pH 7.5, 300 mM NaCl and 0.1 mM phenylmethane sulfonyl fluoride) and lysed by sonication. After centrifugation at 15,000 $g$ for 1 h, the supernatant was loaded onto an Ni-NTA affinity column and washed extensively with lysis buffer. The target protein was eluted using elution buffer (50 mM Tris-HCl pH 7.5, 300 mM NaCl and 300 mM imidazole) and cleaved with PreScission protease overnight at 4 °C. The protein was applied to a HiTrap SP column (GE Healthcare) equilibrated with 50 mM Tris-HCl pH 7.5,

50 mM NaCl, 1 mM DTT and 1 mM EGTA, and eluted by increasing the NaCl concentration. Target protein was collected and further purified by using a Superdex Peptide 10/300 GL column (GE Healthcare). Protein concentrations were determined using a BCA assay kit (Pierce). Mutants were expressed and purified by following the protocol for STIM1 (363–416).

For the $^{15}$N- or $^{13}$C-labeled protein, protein was expressed and purified by following the same protocol used for the unlabeled protein, whereas the Lysogeny broth (LB) medium was replaced with M9 medium supplemented with $^{15}$N-NH$_4$Cl or $^{13}$C-glucose.

**CaM pull-down assay**. Transfected cells were washed twice with Tris-buffered saline (TBS) and lysed in chilled buffer containing 20 mM Tris-HCl (pH 7.5), 150 mM NaCl, 1% Triton X-100, protease inhibitor cocktail (Roche) and 2 mM CaCl$_2$ or 1 mM EGTA. Cell lysates were centrifuged at 15,000 $g$ for 30 min at 4 °C. The supernatant was quantified with a BCA assay kit (Pierce). One milligram of cell lysate (1 mg ml$^{-1}$) was incubated with 30 μl prewashed CAM-Sepharose 4B beads (GE Healthcare) for 2 h in 4 °C. The beads were washed six times with lysis buffer and eluted in 30 μl gel-loading buffer by heating at 95 °C for 10 min. Then, 10 μl of the elution was loaded onto 12–15% SDS-polyacrylamide gel electrophoresis (PAGE) gels or 10–16% tricine–SDS-PAGE (for small molecular weight protein)[63] and transferred to a polyvinylidene difluoride (PVDF) membrane for western blot analysis.

**Immunoprecipitation and western blotting**. Transfected cells were washed with TBS twice and lysed in chilled buffer containing 20 mM Tris-HCl (pH 7.5), 150 mM NaCl, 0.5% Triton X-100, protease inhibitor cocktail (Roche) and 2 mM CaCl$_2$ or 1 mM EGTA. For thapsigargin treatment, cells were treated with 1 μM thapsigargin before lysis for 10 min. Cell lysates were centrifuged at 15,000 $g$ for 30 min at 4 °C. The supernatant was quantified with a BCA assay kit (Pierce). Next, 0.5 mg cell lysate (1 mg ml$^{-1}$) was incubated with rabbit c-Myc-agarose (20 μl slurry, Sigma-Aldrich) or GFP-Trap (15 μl slurry, ChromoTek) for 2–4 h in 4 °C. The beads were washed three times with lysis buffer and eluted in 40 μl gel-loading buffer by heating at 95 °C for 10 min. Then, 15 μl of the eluate was loaded onto 8–12% SDS-PAGE or 10–16% tricine–SDS-PAGE gels and transferred to a PVDF membrane for western analysis. Proteins were immunoblotted using anti-myc (Sigma-Aldrich M5546, 1:7,000), anti-GFP (Abcam ab6556, 1:2,000) and anti-tubulin (Sigma-Aldrich T8203, 1:5,000). The protein–antibody complexes were detected by chemiluminescence. Uncropped original scans of blots are shown in Supplementary Fig. 9.

**Live-cell imaging**. Cells were plated on glass-bottom dishes and transfected at 50% confluence. Cells were imaged in Ringer's solution. Thapsigargin (1 μM; Sigma-Aldrich) was used for store depletion. Enhanced CFP (ECFP), enhanced GFP (EGFP), enhanced YFP (EYFP) and mCherry were excited at 458, 488, 514 and 594 nm, respectively. Fluorescence emission was collected at 465–505 nm for ECFP, 500–540 nm for EGFP, 520–560 nm for EYFP and 605–650 nm for mCherry. All images were captured at room temperature with a ×63 and 1.40 numerical aperture oil-immersion objective lens controlled by LAS software on a Leica SP8 confocal microscope.

**Intracellular Ca²⁺ measurement**. HEK293T was plated on glass-bottom dishes coated with poly-L-lysine (Sigma-Aldrich) and transfected at 60% confluence. Transfected cells were loaded with 2 μM Fura-2 AM (Invitrogen) in Ringer solution at room temperature for 30 min. After loading, cells were transferred to Fura-2 AM-free solution for 30 min. The Ringer solution contained (in mM): 140 NaCl, 5 KCl, 1 MgCl$_2$, 10 D-glucose and 20 HEPES-NaOH, pH 7.4. Fluorescence imaging was undertaken at room temperature using a Leica DMI6000B microscope with a ×63 and 1.40 numerical aperture oil-immersion objective lens controlled by LAS software. Consecutive excitation occurred at 340 and 380 nm every 2 s, and emission was collected at 510 nm. Intracellular Ca²⁺ concentration is shown as the 340/380 ratio. Data are shown as the mean ± SEM.

**FRET measurements**. HEK293T or HEK Orai1-CFP were plated on glass-bottom dishes and transfected with ECFP (donor)- and EYFP (acceptor)-tagged constructs at 50% confluence. Imaging was performed at room temperature using a Leica DMI6000B microscope equipped with high-speed fluorescence-external filter wheels for a CFP-YFP index-based FRET experiment. Images were collected every 10 s using a ×63 oil objective (N.A.1.4; Leica) and processed using the Biosensor Processing Software 2.1 package in MATLAB R2014a. FRET was determined by pixel-by-pixel calculation using apparent FRET efficiency calculation methods[64]. Briefly, bleed-through and direct excitation containment were removed according to the equation $F_c = I_{DA} - aI_{AA} - dI_{DD}$ (assuming $b = c = 0$). The bleed-through factor "$d$" was determined for cells expressing ECFP alone ($d = 0.715$, 92 cells). The direct excitation factor "$a$" was determined for cells expressing EYFP alone ($a = 0.173$, 109 cells). The apparent FRET efficiency (Eapp) was calculated by the equation $E_{app} = F_c/(F_c + GI_{DD})$, where Eapp represents the fraction of donor (ECFP) exhibiting FRET. The microscope-specific constant G was determined by using the D1ER cameleon ($G = 3.458$, 57 cells).

**Solution NMR spectroscopy**. All NMR spectra were recorded at 298 K with a Bruker Avance 600 MHz NMR spectrometer equipped with a QCI-cryoprobe. Protein samples were in 20 mM MES and 5 mM CaCl$_2$ at pH 6.5 unless otherwise indicated. The NMR data were processed with Topspin 2.1 and chemical shifts were assigned with the Sparky and Cara packages. The protein backbone assignments were completed with 3D CBCANH and CBCA(CO)NH spectra.

The interaction between CaM and STIM1 was analyzed by $^{15}$N-HSQC spectra. The binding experiment was performed with $^{15}$N-HSQC spectra of $^{15}$N-STIM1 titrated by CaM. Cross-peak intensity of STIM1 was compared in the presence or absence of CaM. The chemical shift perturbations of STIM1 were plotted with CaM and the dissociation constant ($K_d$) was simulated with Origin.

**Electrophysiological measurements**. HEK293 cells were transiently transfected with CFP-Orai1 and SOAR-YFP, STIM1-YFP or the STIM1-YFP mutant in a ratio of 1:2 for 12–24 h at 37 °C with 5% CO$_2$. After trypsin digestion, transfected cells were plated onto 35 mm dishes with culture medium for at least 1 h to attach cells for electrophysiology. Experiments were performed at room temperature by using the tight-seal whole-cell configuration with the following voltage protocol: 50 ms voltage step to −100 mV from a holding potential of 0 mV, followed by a voltage ramp increasing from −100 to +100 mV in 50 ms with a frequency of 0.5 Hz. To measure the SCDI, we used a pipette solution with 135 mM Cs aspartate, 8 mM MgCl$_2$, 1.2 mM EGTA and 10 mM Cs-HEPES with CaM or its mutant or inhibitors (pH 7.2). The extracellular 0 Ca²⁺ solutions contained the following: 130 mM NaCl, 4.5 mM KCl, 22 mM MgCl$_2$, 10 mM TEA-Cl, 10 mM D-glucose and 5 mM Na-HEPES (pH 7.4). The extracellular Ca²⁺-containing solution contained the following: 130 mM NaCl, 4.5 mM KCl, 20 mM MgCl$_2$, 2 mM CaCl$_2$, 10 mM TEA-Cl, 10 mM D-glucose and 5 mM Na-HEPES (pH 7.4). After whole-cell configuration in the 0 Ca²⁺ extracellular solution, the first whole-cell currents that were recorded were denoted leak currents. Then, the 0 Ca²⁺ extracellular solution was exchanged slowly with 2 mM Ca²⁺ extracellular solution via a perfusion system to record Orai1 currents. Whole-cell currents were amplified with an Axopatch 700B and digitized with a Digidata 1550A system (Molecular Devices). All currents were low-pass filtered at 2 kHz and sampled at 10 kHz. pCLAMP software (Molecular Devices) was used for data acquisition and analysis. To reflect the SCDI process, the current was normalized as follows: the peak current density (pA pF$^{-1}$) was set as −1.0 and the current densities during the measurement were displayed as the ratio of the peak current density to get normalized current densities and were plotted as a function of the recording time. Data points represent the mean ± SEM.

**Competitive binding assay**. Transfected cells were washed twice with TBS and lysed in chilled buffer containing 20 mM Tris-HCl (pH 7.5), 150 mM NaCl, 0.5% NP-40, protease inhibitor cocktail (Roche) and 2 mM CaCl$_2$ or 1 mM EGTA. For the STIM1–Orai1 and STIM1–STIM1 complexes, 3 μM ionomycin was used to induce Ca²⁺ store depletion before lysis. Cell lysates were centrifuged at 15,000 $g$ for 30 min at 4 °C. The complex was pooled with 40 μl GFP-Trap (ChromoTek) for 2 h in 4 °C. After samples were washed twice with lysis buffer, 0.3 mM Ca²⁺ + CaM or EGTA + CaM was added to the complex for binding for 4 h in 4 °C. The beads were washed three times with lysis buffer and eluted in 40 μl gel-loading buffer by heating at 95 °C for 10 min. Then, 10 μl of the elution was loaded onto 7–15% SDS-PAGE gels or 10–16% tricine–SDS-PAGE (for low molecular-weight protein)[63] and transferred to a PVDF membrane for western blot analysis.

**Isothermal titration calorimetry**. Measurements were performed at 16 °C using an ITC-200 microcalorimeter (GE Healthcare). Samples were buffered with 25 mM Tris-HCl (pH 7.5) containing 200 mM NaCl and 2 mM CaCl$_2$ or 1 mM EGTA. To determine the binding affinity of SOAR and CaM, 500 μM CaM or CaM-4EF was injected into 50 μM SOAR. Data were analyzed in the Microcal Origin software.

**Circular dichroism spectroscopy**. Protein secondary structure was recorded in a MOS-450 CD spectrometer (BioLogic) equipped with a temperature controller. Spectral data were measurement at 22 °C using a 1 mm path length quartz cell with 10 μM protein between wavelength of 190–260 nm and using buffer for blank correction.

**Multi-angle light scattering**. Multi-angle light scattering was performed in-line with SEC using a DAWN HELLOS α instrument equipped with an Optilab Refractive Index Detector (Wyatt Technology). Protein (100 μM) was injected into a Superdex 200 10/300 column (GE Healthcare) equilibrated with 25 mM Tris-HCl pH 7.5, 200 mM NaCl at a flow rate of 0.5 ml min$^{-1}$. Protein concentration was calculated from the absorbance at 280 nm and the light scattering data were collected at 663 nm. Molecular weight was calculated using the ASTRA software.

**Statistical analysis**. Statistical significance was determined using unpaired Student's $t$-test with SPSS (IBM) software. No specific randomization or blinding protocols were used. Results are expressed as the mean ± SEM of at least three independent experiments. Differences with a $P$-value of < 0.05 were considered significant: *$P < 0.05$, **$P < 0.01$ and ***$P < 0.001$.

**Data availability**. The authors declare that the data supporting the findings of this study can be found within the paper and its Supplementary Information files, and are available from the corresponding author upon reasonable request.

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

## Acknowledgements

We are grateful to Dr Roger Y. Tsien for providing the Cameleon constructs; to Dr Youjun Wang for providing the HEK Orai1-CFP-stable cells and instructive advices for electrophysiology experiment; to Dr Bingjun He and Dr Qiang Zhao for help in patch clamp experiment; to Dr Xiangyu Wang and Dr Xiangyu Cai for help in protein purification; to Mrs. Shuang Guo, Mrs Xiangchen Guan and Mrs Nannan Xiao for excellent technique service. This work was supported by the 973 Program (grants 2017YFA0504801 and 2013CB910400 to Y.S.), the Natural Science Foundation of China (grants 31370826 and 31570750 to Y.S.; 31300628 to X.Y.; 21473095 to X.S.).

## Author contributions

X.L. performed biochemistry, cell biology and molecular imaging experiments; G.W. performed electrophysiology experiment; Y.Y. and X.S. performed NMR experiment; S.F., X.L., H.K. and X.Y. provided reagents and performed experiments; X.L. and Y.S. designed the experiments and wrote the paper.

## Additional information

**Competing interests:** The authors declare no competing financial interests.

