## [Peer review file · Nature Communications]

Reviewers' comments:

Reviewer #1 (Remarks to the Author):

The manuscript of Li et al. demonstrates Ca²⁺ dependent CaM binding to STIM1/SOAR, which disrupts the STIM1/SOAR-Orai1 complex as well as inhibits STIM1 oligomerization. The results are interesting and timely. The experiments appear carefully conducted and most of the conclusions drawn are justified. I would like the authors to address the points given below to strengthen their manuscript.

- 1) Electrophysiological experiments with SOAR+ Orai1 as shown for example in Fig. 2A are expected to yield constitutive currents at the very beginning. However, here it looks almost similar to STIM1-dependent currents activated via store depletion?
- 2) The authors observe enhanced SCDI upon infusion of CaM via the pipette in Fig. 2. Effects on FRET have been shown with overexpression of CaM in e.g. Fig. 4G. Thus, how do the effects look like in electrophysiological experiments when you overexpress CaM?
- 3) As the effects are sometimes quite small as e.g. in Fig. 2C or Fig. 2G, the standard deviation or SEM values are missing to evaluate for a significant difference.
- 4) All experiments in Fig. 2 are normalized. It could be, however, that the extent of SCDI is dependent on the current size in a sense that larger currents show enhanced SCDI. Is there any correlation seen?
- 5) With respect to the mutations L390S/F391S that assumedly has several inhibitory effects, I would be interested to see to which extent single point mutation of either residue work.
- 6) Does the region around L390/F391 in SOAR represent a typical CaM binding motif or how can it be excluded that this involves an allosteric mechanism?
- 7) Fig 3D: The authors measured an average KD via three residues A369, G379 and I383. Why didn't the authors compare directly L390 and F391 with L390S-F391S?
- 8) Although mechanistically not fully clear, it has been shown that substitutions of W76 in the Orai1 N-terminus with negatively charged residues abolishes fast Ca²⁺ dependent inactivation (FCDI). Thus, it would be interesting to see whether SCDI as shown here occurs independent of FCDI. In particular, this may help to exclude that the presented CaM-dependent functional effects additionally involve a potential Orai1-CaM-binding site?

Minor points

- 9) Fig 1 G: Is CaM binding to SOAR significantly different to that of CaM-4EF mutants? It seems that the Eapp SEM-bars of the two binding curves overlap in addition to that they are pretty small. It might be interesting to check for a binding of CaM or CaM-4EF to SOAR, when cytosolic Ca²⁺ concentration is increased via ionomycin, yet in the absence of Orai1.
- 10) Fig 2: The I/V curves in Figure 2 (A-D), all possess a rather low reversal potential compared to the typical reversal potential of +50mV of CRAC or STIM1/Orai1 currents. Additionally, in Fig. 2B the reversal potentials seem to be different in the absence and presence of CaM-4EF. Generally, are the depicted currents or I/V curves leak corrected?

Reviewer #2 (Remarks to the Author):

The authors present a potentially interesting paper on the interaction of calmodulin (CaM) with the STIM1-Orai1 complex and its effects to dissociate the oligomeric complex of STIM1 and the slow inactivation of the Orai1 channel. There has been much work undertaken on the control of STIM1-Orai1 and the actions of CaM. The authors have made some major contributions to understanding the structure of STIM1 and how it may interact with and activate the Orai1 channel. The current paper does add some new perspective to this area but unfortunately the work has some significant shortcomings and inconsistencies which are detailed below.

The manuscript addresses a CaM-mediated slow Ca²⁺ dependent inactivation (CDI) mechanism that controls store-operated Ca²⁺ entry (SOCE). Several earlier studies have focused on the relationship between CaM and the Orai channel in CDI. The role of CaM on fast CDI is still a controversial issue. Based on results from ITC, IP, patch-clamp, FRET and confocal imaging, the authors propose a new role for CaM in which CaM binds to the core region of activated STIM1 and assists slow CDI, as opposed to altering fast CDI through binding to Orai1. This work provides in vitro evidence for the interaction between CaM and STIM1/ SOAR, which also affects the interaction between STIM1 and Orai1 and leads to deactivation of CRAC channel. The authors also identified two amino acids, L390 and F391 within the SOAR fragment which appear to play a critical role for the CaM binding to STIM1. However, the mechanistic conclusions of the paper on physiological STIM1-Orai1 coupling process are very unclear. Some of the work, particularly the patch-clamp data, appear to have significant flaws and are difficult to understand or interpret. STIM1 mutants that supposedly alter the CaM binding domain of STIM1, prevent STIM1 and SOAR from binding to Orai1 and prevent STIM1 aggregation, however, the connection to slow inactivation is unclear. Also, proof of the CaM-binding region from the LF/SS mutation and short peptides is not substantial. Together, the flaws in the paper are very significant and would be difficult to reconcile.

The following are some comments reflecting major problems of the paper.

(1) The data in Fig 2 (and suppl. Fig 1) are troubling. First, the I/V curves do not match the current traces over time (e.g. panels B and C). Second, and more troubling, with SOAR expression the current should be constitutive and already fully activated upon break in. The fact that it is not, suggests that these traces do not correspond to the function of expressed SOAR. They look like store-operated currents. If only one trace was like this one might think it was a mistake, but with all the traces for SOAR showing this aberrant behaviour, it suggest a serious flaw in the experimental approach.

(2) In addition to this problem, the patch clamp data in Fig. 2 and supplement are not quantified statistically to document the effects on slow inactivation which is the purpose of this study.

(3) Although the knockdown of CaM might be rather difficult, inferring the involvement of CaM from the action of W7 is not acceptable. The W7 compound is notoriously non-specific. Moreover, the TRP peptide, even if established to bind specifically to TRP, proves very little since it still can have confounding non-specific effects. Furthermore, CaM binding to TRPs still remains a contentious issue.

(4) In three figures (Figs. 4-6), the authors' mutagenesis and FRET/Co-IP/Ca²⁺ imaging data suggest that L390 and F391 are important in SOAR-Orai1 and STIM1-Orai1 binding, STIM1 aggregation and Orai1 channel function. This is certainly expected given the location of these residues which are immediately adjacent to residues (e.g. F394) that are established to constitute the Orai1-binding site. The Fig.3 data show weak binding of the mutant compared to WT in vitro and in overexpression but within the context of a short peptide or SOAR region for the co-IP. However, these data do not constitute any proof that these residues are required for Ca²⁺/CaM binding in vivo under physiological conditions.

(5) In Fig 1, the pulldown data show that STIM1-c2 (1-444) but not c3 and c4 interact, prompting the conclusion that only the exposed SOAR is necessary for the association of C-terminal STIM1 with Ca²⁺/CaM. This conclusion is not supported by data.

The following are other significant problems with the paper:

(6) The figure legends miss some key information. In Fig. 1C -1F, it is not clear whether Orai1 is overexpressed. Since the presence of CaM will affect the interaction between STIM1/SOAR and Orai1, it is reasonable to assume that Orai1 might be affected by the binding between

STIM1/SOAR and CaM. Thus, it is important to point this out in these figure panels.

(7) In a similar issue as above, in Fig. 2, the addition of Ca²⁺ needs to be labeled. Also, it is not clear where the IV curves come from, the peak, the plateau, or any time after peak? The IV curves are current density (pF/pA), while the time courses are normalized current, so this is really confusing. Moreover, the authors need to explain how the current was normalized. It seems that the current is not normalized to fluorescence intensity. So what is the justification for normalization?

(8) In Fig. 2 and Supplemental Figure 1, the time course of graphs and related I/V curves have conflicting information. In Fig. 2B, the I/V curve shows 100 μ M CaM-4EF increase the current from 10 to 18 pA/pF, while the time course graph shows no effect. The same problem shows up in Fig. 2C, Fig. 2E-2G and Supplemental Figure 1A-C.

(9) In the methods, only one pipette solution recipe was provided. If the authors use the same pipette solution for all the figures, then the Fig. 2 data are questionable. 1.2mM EGTA alone in the pipette solution can barely generate CRAC current. Thus the method for the patch-clamp experiments is incomplete. The authors need to make a clear statement about what recipe is used for each figure. More importantly, as stated above, the Fig. 2A/2B data for SOAR should reveal constitutively active current, while Figure 2C, 2D, 2F and 2G for STIM1 show the characteristic slow development of CRAC current that is expected. So it is impossible to understand why the SOAR data also reaches a peak after the same lag of about 50 seconds. This indicates the experiment and all interpretation is totally flawed.

(10) On page 5 (L13), the authors provide evidence (Figure 1A) that the STIM1 truncation which perturbed SOAR cannot associate with CaM (such as STIM1 (268-420)). However, in Fig. 3B and D, the authors claim that the interaction between STIM1 (363-416) and CaM exists and can be detected. These contradictory results/statements are not discussed and not explained.

(11) In a previous report (Qi et al., 2009), 1 μ M W7 was enough to inhibit CaM, so what is the justification for using 30/100 μ M here? Why not use 1 μ M W7? The concentration used here is 30 or 100 times higher than reported. As stated above, W7 is very nonspecific to this high level makes the effect invalid. Also, in Fig. 2F, only 100 μ M W7 showed some effect. Again, this effect of W7 does not allow a conclusion. The better control for Fig. 2 is to add 20 mM BAPTA and 100 μ M CaM in the pipette solution which is more specific for the inhibition of CaM.

(12) The FRET data have the same problems as the current data – consistency. In Fig. 5C normalized Eapp was used, while Eapp was used in Fig. 1G and Fig. 6C.

(13) The ratio of donor/acceptor plays a critical role in determining the final Eapp value. To make an accurate and reliable conclusion, the donor/acceptor ratio for the two compared groups need to be provided and need to be in a similar narrow range. Otherwise, even in the control group, the Eapp will vary a lot. In Fig. 4, the FRET between WT SOAR and WT Orai1 is about 0.3 Fig. 4B, but it is about 0.24 in Fig. 4G and 4H. This is about a 25% difference. However, in Fig. 4G and 4H, the treatment resulted in about 10% change compared to control and the authors considered this as significant difference. If the variation in the control group is above 20%, it is hard to believe the 10% difference between control group and treated group is significant. Thus, the ratio of donor/acceptor in each individual group is a must. Only cells with similar donor/acceptor ratio are meaningful. The same issue applies for for all other FRET data, such as Fig. 1G, the peak FRET value for CaM-SOAR is about 0.03, which is close to the background FRET value for Orai1/SOAR-LF/SS that is about 0.025. Also, the error bar in this figure is huge.

(14) On page 11, line 4 from bottom, "After 5 minutes of extracellular Ca²⁺ influx, the FRET Eapp value at the PM was significantly higher in W-7-treated cells than in DMSO-treated cells (Figure 4H)". It would definitely have been more convincing if the FRET value at time 0 was provided. This

is an important criticism.

Reviewer #3 (Remarks to the Author):

The manuscript (NCOMMS-17-05452-T) presents a wealth of structural and functional data (both in vitro and in vivo) to convincingly demonstrate that Ca²⁺-bound CaM (Ca/CaM) binds to the protein Stim1 and promotes Ca²⁺-induced inactivation of the Orai channel, which is important for Store-operated Ca²⁺ entry (SOCE). This work is potentially significant to the field and should be published. Overall, the functional and structural data and analysis all appear satisfactory. However, I have a few comments and questions below about the NMR structural analysis that should be addressed before a final recommendation can be made.

1. Supplemental Figure 2 shows a very nice HSQC spectrum of ¹⁵N-labeled STIM1 (363-416) construct, and a titration of this spectrum with Ca/CaM reveals key residues of STIM1 (A369, G379 and I383) that likely make contact to Ca/CaM (Fig. 3C). It would be good to also show (in Fig S2) an overlay of the HSQC spectrum of ¹⁵N-labeled STIM1 (363-416) construct in the presence of saturating unlabeled Ca/CaM to better illustrate which residues are affected the most upon Ca/CaM binding. In other words, Fig S2 should show spectrum of STIM1 alone in black on top of a spectrum of STIM1 in presence CaM in red to illustrate which STIM1 residues are most affected by CaM binding.
2. In a separate supplemental figure, I recommend showing HSQC spectrum of ¹⁵N-labeled Ca/CaM in the presence of unlabeled STIM1(363-416). This spectrum would then identify the residues in CaM that are most affected by STIM1 binding. The chemical shift perturbation data for both CaM and STIM1 could then be used to guide the docking of the known structures of CaM and STIM1(363-416) solved previously by Yang et al in Proc Natl Acad Sci U S A. 2012 Apr 10;109(15):5657-62. An atomic-level structural model of the CaM/STIM1 complex would be a nice addition to this paper and would not require that much extra work.
3. Fig. 3c and 3d show NMR titration of Ca/CaM binding to the ¹⁵N-STIM1 (363-416) mutant L390S-F391S. The K_d values determined for the different residues seem to be rather different (138 μM for A369 vs 71 μM for I383). What is the explanation/interpretation for the different K_d values here? Do the different K_d values indicate two different sites? The ITC binding data in Fig. 1B indicate there is only one site. So, it is not clear why two different K_d values are measured for the two residues. There needs to be a more detailed molecular interpretation of the different K_d values here and what is the functional significance?
4. For the NMR titration in Figs. 3c and 3d, it would be good to also show the corresponding NMR titration data for wild type STIM1 binding to CaM. Does the NMR titration for wild type show fast exchange (ie single peak that moves like in Fig. 3c) or is it slow exchange (ie two separate peaks) due to the higher binding affinity?
5. Does the STIM1 SOAR region (residues 363-416) contain a known calmodulin binding motif? If so, what type of CaM motif is it (IQ, 1-12 etc)? There needs to be more discussion about the residues in STIM1 that interact with CaM, and relate the amino acid sequence of CaM binding site in STIM1 with other known CaM binding motifs.

Reviewers' comments:

Reviewer #1 (*Remarks to the Author*):

The manuscript of Li et al. demonstrates Ca²⁺ dependent CaM binding to STIM1/SOAR, which disrupts the STIM1/SOAR-Orai1 complex as well as inhibits STIM1 oligomerization. The results are interesting and timely. The experiments appear carefully conducted and most of the conclusions drawn are justified. I would like the authors to address the points given below to strengthen their manuscript.

1) Electrophysiological experiments with SOAR+ Orai1 as shown for example in Fig. 2A are expected to yield constitutive currents at the very beginning. However, here it looks almost similar to STIM1-dependent currents activated via store depletion?

Response: To investigate the SCDI of Orai1, long-term and stable current recording were essential. However, in our system, long-term current recording was challenging due to the instability of the high-quality seal if extracellular Ca²⁺ was immediately added to the cell surface. Therefore, we used the following protocol to record the currents in our experiments that are shown in **Fig. 2** of our revised manuscript.

First, whole-cell configuration in the 0 Ca²⁺ extracellular solution was obtained and the whole-cell currents were then recorded as the leak currents. Next, the 0 Ca²⁺ extracellular solution was exchanged slowly via perfusion system with 2 mM Ca²⁺ extracellular solution to record currents. This recording protocol can make sure that a high-quality seal is present and obtain long-term and stable currents. We have included this detailed protocol in the **Methods** section **Electrophysiological measurements** in our revised manuscript.

In this way, Ca²⁺ was slowly diffused to the cell surface. The net Orai1 channel currents at the very beginning that were mediated by Ca²⁺ were almost negligible after correction for the leak currents (see the following Fig., **left side**). Therefore, at the very beginning, SOAR+Orai1 induced currents that are similar to the currents that STIM1+Orai1 induced currents. Our results are consistent with the data (Figs. 2C, 5C) in a previous paper (**Jha et al. J. Cell Biol., 202: 71-79, 2013**). Additionally, we also measured the currents using HKE293 cells that expressed SOAR+Orai1 when 2 mM extracellular Ca²⁺ was immediately added into the cell surface (see the following Fig., **right side**). We found that the constitutive currents were recorded at the very beginning. However, as shown in the figure, the current

recording lasted only 100 seconds and not longer.

2) The authors observe enhanced SCDI upon infusion of CaM via the pipette in Fig.2. Effects on FRET have been shown with overexpression of CaM in e.g. Fig. 4G. Thus, how do the effects look like in electrophysiological experiments when you overexpress CaM?

Response: In whole-cell patch clamp mode, the cell membrane that was patched under the borosilicate glass pipette must be broken to obtain the whole-cell configuration so that the pipette solution can contact the intracellular solutions. Under this circumstance, the cytosolic CaM inside the cell will be quickly washed out into the pipette solutions, resulting in a substantial drop of cytosolic CaM concentration to a negligible quantity. However, in FRET experiments, the cell membrane integrity will be kept, and overexpressing CaM inside the cell can result in a higher concentration of cytosolic CaM in these cells than in control cells. This reason is why we chose to use a perfusion system to increase the CaM level when recording SCDI currents, while overexpressing CaM was used in the FRET experiments.

3) As the effects are sometimes quite small as e.g. in Fig. 2C or Fig. 2G, the standard deviation or SEM values are missing to evaluate for a significant difference.

Response: As suggested by the reviewer, the error bar (SEM values) was added to **Fig. 2** of the revised manuscript.

4) All experiments in Fig. 2 are normalized. It could be, however, that the extent of

SCDI is dependent on the current size in a sense that larger currents show enhanced SCDI. Is there any correlation seen?

Response: We thank the reviewer for the suggestion. We have carefully gone through our patch-clamp data and did not find any correlation between the extent of SCDI and the current magnitude.

5) With respect to the mutations L390S/F391S that assumedly has several inhibitory effects, I would be interested to see to which extent single point mutation of either residue work.

Response: As suggested by the reviewer, we performed FRET experiments to test the effect of a single mutation (L390S or F391S) on STIM1-Orai1 binding. We found that a single mutation decreased the STIM1-Orai1 FRET value to a similar extent and double mutation almost abolished STIM1-Orai1 interaction (see the following Fig., **left side**), indicating that both residues contribute to the STIM1-Orai1 interaction. We also performed confocal imaging and showed that a single mutation was not able to abolish the STIM1-Orai1 interaction after TG stimulation (see the following Fig., **right side**), while double mutation abrogated the interaction (**Figs. 5a, 6b**). These results indicate that both residues contribute to STIM1-Orai1 binding. Moreover, our modeling studies (**Supplementary Fig. 5**) revealed that these two residues are deeply buried inside the C-lobe of CaM.

6) Does the region around L390/F391 in SOAR represent a typical CaM binding motif or how can it be excluded that this involves an allosteric mechanism?

Response: To the best of our knowledge, the region around L390/F391 in SOAR does not represent a canonical CaM-binding motif. We further analyzed the CaM-binding site in SOAR using the Calmodulin Target Database (Yap et al., *J. Struct. Funct. Genomics.*, **1(1): 8-14, 2000**). The region near L390/F391 represents the most likely binding site in SOAR. Our NMR titration experiments confirmed that these two residues are involved in the association with CaM. Moreover, ^{15}N -STIM1 and ^{15}N -STIM1-L390S/F391S produced similar ^{15}N -HSQC spectra (**Supplementary Fig. 8**), indicating that the L390/F391 mutations did not change the secondary structure of SOAR. In addition, multiple pieces of biochemical data showed that L390 and F391 are not involved in SOAR dimerization and structural stability (**Supplementary Fig. 7**). Our personal biased view is that an allosteric mechanism is unlikely to be involved.

7) Fig 3D: The authors measured an average K_d via three residues A369, G379 and I383. Why didn't the authors compare directly L390 and F391 with L390S-F391S?

Response: In the NMR experiment that involves titrating ^{15}N -STIM1 (363-416) wild-type with CaM/ Ca^{2+} , the peak intensities of residues L390 and F391 disappeared very quickly (**Supplementary Fig. 3b** of the revised manuscript) and thus are not suitable to be used for determining K_d . Instead, the peak intensities of the residues that gradually changed were selected for analysis. In the NMR experiment in which the ^{15}N -STIM1 (363-416) L390S-F391S mutant was titrated with CaM/ Ca^{2+} , three residues, A369, G379 and I383, showed nice peak intensity shifts as the amount of CaM increased (**Figs. 3c,d** of the revised manuscript). The K_d that were determined using the three residues were averaged to reduce the error. The dissociation constants that were determined by the chemical shift changes varied among different residues, perhaps reflecting the variations in local conformational changes during the association of STIM1 (363-416) with Ca^{2+} /CaM.

8) Although mechanistically not fully clear, it has been shown that substitutions of W76 in the Orail N-terminus with negatively charged residues abolishes fast Ca^{2+} -dependent inactivation (FCDI). Thus, it would be interesting to see whether SCDI as shown here occurs independent of FCDI. In particular, this may help to exclude that the presented CaM-dependent functional effects additionally involve a potential Orail-CaM-binding site?

Response: As suggested by the reviewer, we performed electrophysiological experiments and FRET experiments to address whether CaM-dependent SCDI is relevant to FCDI. First, we tested the extent that SCDI was regulated by CaM by using the Orail-W76E mutant. Our results showed that CaM indeed increases the extent of SCDI in the Orail-W76E channel (see the following Fig., left side). It was reported that the Orail-W76E mutant did not exhibit FCDI (Mullins et al., *Proc. Natl. Acad. Sci.*, **106(36): 15495-500, 2009**). Second, we measured SOAR-CaM FRET in cells that co-expressed the Orail-W76E mutant after calcium influx. Our results showed that after calcium influx, the Orail-W76E mutant did not interfere with the SOAR-CaM interaction (see the following Fig., right side). These results indicate that CaM-dependent SCDI is most likely independent of FCDI.

Minor points

9) Fig 1 G: Is CaM binding to SOAR significantly different to that of CaM-4EF mutants? It seems that the E_{app} SEM-bars of the two binding curves overlap in addition to that they are pretty small. It might be interesting to check for a binding of CaM or CaM-4EF to SOAR, when cytosolic Ca^{2+} concentration is increased via ionomycin, yet in the absence of Orail.

Response: As suggested by the reviewer, we performed the FRET experiment in which CaM or the CaM-4EF mutant bound to SOAR in the absence of Orai1. Our results showed that the FRET value between SOAR and CaM is slightly increased via ionomycin, while FRET values are similar between SOAR and CaM-4EF (see the following Fig., **left side**). We also found that the extent of the FRET increase via ionomycin treatment is relatively small in the absence of Orai1 compared with the presence of Orai1 (see the following Fig., **right side**). One possibility is that the SOAR-Orai1 coupling may promote SOAR to a conformation that is feasible for the approach of Ca²⁺/CaM.

As noted by the reviewer, the peak E_{app} value for SOAR-CaM is indeed relatively small, which may be due to the large amount of endogenous CaM. The endogenous CaM will compete with overexpressed CaM-CFP to bind with SOAR-YFP and thus decrease the FRET signal between CFP and YFP.

10) Fig 2: The I/V curves in Figure 2 (A-D), all possess a rather low reversal potential compared to the typical reversal potential of +50mV of CRAC or STIM1/Orai1 currents. Additionally, in Fig. 2B the reversal potentials seem to be different in the absence and presence of CaM-4EF. Generally, are the depicted currents or I/V curves leak corrected?

Response: We carefully went through our data and confirmed that the currents in our manuscript were leak corrected. In our system, we used a low extracellular Ca²⁺ concentration (2 mM) in the bath and a low EGTA concentration (1.2 mM) in the pipette to record SCDI, which may lead to a rather low reversal potential. However, in a previous report (Prakriya et al. *Nature*, **443**, 230–233, 2006), the typical reversal potential of +50 mV of CRAC or STIM1/Orai1 currents was recorded under the

conditions of higher extracellular Ca^{2+} (10-20 mM) in the bath and higher EGTA concentration (10-20 mM) in the pipette.

We thank the reviewer for pointing out the problem that the reversal potentials seem to be different in the absence and presence of CaM-4EF. At the current stage, we have no idea why the reversal potentials are different in the absence and presence of CaM-4EF. Since this point is not closely related to our current manuscript, we intend to leave this question open for future studies.

Reviewer #2 (Remarks to the Author):

The authors present a potentially interesting paper on the interaction of calmodulin (CaM) with the STIM1-Orai1 complex and its effects to dissociate the oligomeric complex of STIM1 and the slow inactivation of the Orai1 channel. There has been much work undertaken on the control of STIM1-Orai1 and the actions of CaM. The authors have made some major contributions to understanding the structure of STIM1 and how it may interact with and activate the Orai1 channel. The current paper does add some new perspective to this area but unfortunately the work has some significant shortcomings and inconsistencies which are detailed below.

The manuscript addresses a CaM-mediated slow Ca^{2+} dependent inactivation (CDI) mechanism that controls store-operated Ca^{2+} entry (SOCE). Several earlier studies have focused on the relationship between CaM and the Orai channel in CDI. The role of CaM on fast CDI is still a controversial issue. Based on results from ITC, IP, patch-clamp, FRET and confocal imaging, the authors propose a new role for CaM in which CaM binds to the core region of activated STIM1 and assists slow CDI, as opposed to altering fast CDI through binding to Orai1. This work provides in vitro evidence for the interaction between CaM and STIM1/ SOAR, which also affects the interaction between STIM1 and Orai1 and leads to deactivation of CRAC channel. The authors also identified two amino acids, L390 and F391 within the SOAR fragment which appear to play a critical role for the CaM binding to STIM1. However, the mechanistic conclusions of the paper on physiological STIM1-Orai1 coupling process are very unclear. Some of the work, particularly the patch-clamp data, appear to have significant flaws and are difficult to understand or interpret. STIM1 mutants that supposedly alter the CaM binding domain of STIM1, prevent STIM1 and SOAR from binding to Orai1 and prevent STIM1 aggregation, however, the connection to slow inactivation is unclear. Also, proof of the CaM-binding region from the LF/SS mutation and short peptides is not substantial. Together, the flaws in the paper are very significant and would be difficult to reconcile.

The following are some comments reflecting major problems of the paper.

(1) The data in Fig 2 (and suppl. Fig 1) are troubling. First, the I/V curves do not match the current traces over time (e.g. panels B and C). Second, and more troubling, with

SOAR expression the current should be constitutive and already fully activated upon break in. The fact that it is not, suggests that these traces do not correspond to the function of expressed SOAR. They look like store-operated currents. If only one trace was like this one might think it was a mistake, but with all the traces for SOAR showing this aberrant behaviour, it suggest a serious flaw in the experimental approach.

Response:

1) The I/V curves represent the current magnitude and had no any correlation with the time course of normalized SCDI. The purpose of showing the I/V curves in our manuscript was to justify that the Orai1 channel currents were truly recorded with inward rectification. In the experiment, the I/V curves and the current traces over time were generated using different methodologies. Briefly, the I/V curves were generated from the peak current induced by the potential ramping protocol, while the current traces over time were generated from the current that was induced by the potential step protocol. Our results were consistent with those of many previous reports (Parekh et al., *J. Biol. Chem.*, **273(24): 14925-14932, 1998**; Jha et al., *J. Cell Biol.*, **202: 71-79, 2013**). To avoid potential confusion, all I/V curves were put into the revised **Supplementary Fig. 1**, and detailed current recording protocols were included in the **Methods** section **Electrophysiological measurements** in the revised manuscript.

2) The current traces are dependent on the current recording protocol. We used the following protocol to record the whole-cell currents that were induced by SOAR+Orai1: First, the whole-cell configuration in the 0 Ca²⁺ extracellular solution was obtained, and the whole-cell currents were then recorded as the leak currents. Next, the 0 Ca²⁺ extracellular solution was exchanged slowly via a perfusion system with 2 mM Ca²⁺ extracellular solution to record currents. This recording protocol can make sure that a high-quality seal is present and can obtain long-term and stable currents. Therefore, the traces for SOAR in our manuscript are correct and consistent with results from many published papers. In our system, maintaining a high-quality seal is troublesome during long SCDI recording times if extracellular Ca²⁺ was immediately added to the cell surface.

(2) *In addition to this problem, the patch clamp data in Fig. 2 and supplement are not quantified statistically to document the effects on slow inactivation which is the purpose of this study.*

Response: As suggested by the reviewer, the error bar (SEM values) was added to **Fig. 2** of the revised manuscript.

(3) *Although the knockdown of CaM might be rather difficult, inferring the involvement of CaM from the action of W7 is not acceptable. The W7 compound is notoriously non-specific. Moreover, the TRP peptide, even if established to bind specifically to TRP, proves very little since it still can have confounding non-specific effects. Furthermore, CaM binding to TRPs still remains a contentious issue.*

Response: The W7 compound and the TRP peptide are canonical CaM inhibitors, and they have been widely used for the purpose of inhibiting CaM function in many previous papers (**Moreau et al, Curr. Biol., 16(16): 1672-7, 2006; Torok et al, Curr. Biol., 8(12):692-9, 1998; Yamashita et al, Nat. Neurosci., 13(7): 838-44, 2010**). Moreover, W7 is a small chemical compound, and TRP is a peptide. They bind to CaM with different modes and consequently inhibit CaM function via different ways. Our results showed that the effect of CaM-dependent SCDI can be inhibited using both inhibitors. Such cross-validated results provide convincing evidence for CaM-dependent SCDI.

(4) *In three figures (Figs. 4-6), the authors' mutagenesis and FRET/Co-IP/Ca²⁺ imaging data suggest that L390 and F391 are important in SOAR-Orai1 and STIM1-Orai1 binding, STIM1 aggregation and Orai1 channel function. This is certainly expected given the location of these residues which are immediately adjacent to residues (e.g. F394) that are established to constitute the Orai1-binding site. The Fig.3 data show weak binding of the mutant compared to WT in vitro and in overexpression but within the context of a short peptide or SOAR region for the co-IP. However, these data do not constitute any proof that these residues are required for Ca²⁺/CaM binding in vivo under physiological conditions.*

Response: As suggested by the reviewer, we performed the co-IP experiment to investigate whether L390 and F391 residues are required for Ca²⁺/CaM binding *in vivo*

under physiological conditions. Our results showed that the co-expression of SOAR wild type and CaM resulted in strong binding between them, while the SOAR-LF/SS mutant did not bind CaM strongly. Furthermore, the STIM1 (1-444) construct that contains the full SOAR domain was used for validation. The wild type but not the LF/SS mutant can interact with CaM after TG treatment (see the following Fig.; this figure was also included as **Supplementary Fig. 6** in the revised manuscript). In conclusion, these data provided evidence that these two residues are required for Ca^{2+} /CaM binding *in vivo*.

(5) In Fig 1, the pulldown data show that STIM1-c2 (1-444) but not c3 and c4 interact, prompting the conclusion that only the exposed SOAR is necessary for the association of C-terminal STIM1 with Ca^{2+} /CaM. This conclusion is not supported by data.

Response: The STIM1 constructs that we used in **Fig. 1a** were C-terminal cytosolic fragments that did not include the N-terminal and transmembrane domains of STIM1. The construct c2 (234-444) contains the entire SOAR and CC1 domains. The CC1 domain interacts with the bottom of V-shaped SOAR but does not cover the upper part of V-shaped SOAR, as reported in the crystal structure of the construct c2 from *Caenorhabditis elegans* (Yang et al., *Proc. Natl. Acad. Sci.*, **109(15): 5657-62, 2012**). In other words, SOAR is exposed in the construct c2 (234-444). The constructs c3 and c4 contain the CTID domain, which is in position to interfere with dimerization of SOAR (Jha et al., *J. Cell Biol.*, **202: 71-79, 2013**); consequently, these two constructs cannot interact with Ca^{2+} /CaM. Therefore, the conclusion “thus suggesting that both

correct SOAR conformation and exposed SOAR are necessary for the association of C-terminal STIM1 with Ca²⁺/CaM.” is reasonable.

The following are other significant problems with the paper:

(6) The figure legends miss some key information. In Fig. 1C -1F, it is not clear whether Orai1 is overexpressed. Since the presence of CaM will affect the interaction between STIM1/SOAR and Orai1, it is reasonable to assume that Orai1 might be affected by the binding between STIM1/SOAR and CaM. Thus, it is important to point this out in these figure panels.

Response: In **Figs. 1c-f**, the Orai1 is not overexpressed. As suggested by the reviewer, the information is included in the figure legend of the revised manuscript.

(7) In a similar issue as above, in Fig. 2, the addition of Ca²⁺ needs to be labeled. Also, it is not clear where the I/V curves come from, the peak, the plateau, or any time after peak? The I/V curves are current density (pF/pA), while the time courses are normalized current, so this is really confusing. Moreover, the authors need to explain how the current was normalized. It seems that the current is not normalized to fluorescence intensity. So what is the justification for normalization?

Response:

- 1) As suggested by the reviewer, the addition of Ca²⁺ was included in the figure legend of **Fig. 2** in our revised manuscript.
- 2) I/V curves were generated from the data points of the peak current that were induced by the ramp stimulation protocol.
- 3) Presenting the current density (pA/pF) I/V curves is commonly employed in Orai1 channel studies, as reported in many previous papers (**Wang et al., Nat. Commun., 5: 3183, 2014; Zhou et al., Nat. Commun., 7: 13725, 2016**). The purpose of showing I/V curves is to justify that Orai1 channel currents were truly recorded with inward rectification. In terms of SCDI measurement, due to the large variation in single-cell current magnitude, the current trace over time must be normalized to be able to make a comparison, as reported in the paper (**Jha et al. J. Cell Biol., 202: 71-79, 2013**).
- 4) The currents were normalized as follows: the peak current was set to -1.0; the currents at any given time were shown as the ratio of the peak currents. This method

of normalization is consistent with that of the previous paper (Jha et al., *J. Cell Biol.*, **202: 71-79, 2013**).

5) The current is not normalized to fluorescence intensity. To the best of our knowledge, the current has no correlation with the fluorescence intensity.

(8) In Fig. 2 and Supplemental Figure 1, the time course of graphs and related I/V curves have conflicting information. In Fig. 2B, the I/V curve shows 100 uM CaM-4EF increase the current from 10 to 18 pA/pF, while the time course graph shows no effect. The same problem shows up in Fig. 2C, Fig. 2E-2G and Supplemental Figure 1A-C.

Response: The I/V curves did not have any correlation with the time course of the normalized SCDI trace. They were generated using different methodologies and served different purposes. The I/V curves were shown to justify that Orai1 channel currents are characteristic of typical inward rectification. The time courses of normalized SCDI traces were shown to demonstrate the extent of Orai1 channel inactivation.

(9) In the methods, only one pipette solution recipe was provided. If the authors use the same pipette solution for all the figures, then the Fig. 2 data are questionable. 1.2mM EGTA alone in the pipette solution can barely generate CRAC current. Thus the method for the patch-clamp experiments is incomplete. The authors need to make a clear statement about what recipe is used for each figure. More importantly, as stated above, the Fig. 2A/2B data for SOAR should reveal constitutively active current, while Figure 2C, 2D, 2F and 2G for STIM1 show the characteristic slow development of CRAC current that is expected. So it is impossible to understand why the SOAR data also reaches a peak after the same lag of about 50 seconds. This indicates the experiment and all interpretation is totally flawed.

Response:

1) One pipette solution recipe was used in our experiment, and the ingredients were included in the **Methods** section **Electrophysiological measurements** in the revised manuscript.

2) The shape of current trace over time depends the protocol used to record the current. In our manuscript, we used the following protocol to record the whole-cell current for SOAR+Orai1: First, the whole-cell configuration in the 0 Ca²⁺

extracellular solution was obtained, and the whole-cell currents were then recorded as the leak currents. Next, the 0 Ca²⁺ extracellular solution was exchanged slowly via a perfusion system with a 2 mM Ca²⁺ extracellular solution to record currents. In this way, there will be a lag before SOAR-induced current reaches its peak. Our results are consistent with the data (**Fig. 5c**) of a previous paper (**Jha et al. J. Cell Biol., 202: 71-79, 2013**).

(10) On page 5 (L13), the authors provide evidence (Figure 1A) that the STIM1 truncation which perturbed SOAR cannot associate with CaM (such as STIM1 (268-420). However, in Fig. 3B and D, the authors claim that the interaction between STIM1 (363-416) and CaM exists and can be detected. These contradictory results/statements are not discussed and not explained.

Response: We used two different methodologies to study the interaction between STIM1 and CaM in **Fig. 1a** and **Fig. 3**. These results are not comparable since their systems are totally different. To the best of our knowledge, the co-IP method that was used for **Fig. 1a** is suitable for detecting strong binding (μM range) in solution when the concentrations of the binding partner are in the nM range. In contrast, the NMR method that was used for **Fig. 3** is suitable for weak binding (mM range) in a system in which the concentrations of the binding partners are in the mM range. Therefore, it is reasonable that there will be some differences in the interactions due to the million-fold concentration difference between the two methods. At the beginning, we tried to use the intact SOAR domain to search for the binding site in an NMR experiment. Unfortunately, the intact SOAR domain aggregated under NMR conditions. After extensive screening, STIM1 (363-416) gave us well-resolved peaks in the NMR spectrum and thus could be used for further NMR studies. Compared with the intact SOAR domain, STIM1 (363-416) showed much weaker affinity with Ca²⁺/CaM, when we used the same ITC methodology, i.e., similar protein concentration, buffer system and signal detection. This K_d value is 27 μM (see the following figure) for SOAR (363-416) with Ca²⁺/CaM, approximately 100-fold weaker than the binding of intact SOAR with Ca²⁺/CaM (0.23 μM in **Fig. 1b**). Nevertheless, three different

methodologies (co-IP, NMR and ITC) resulted in a similar result: that the intact SOAR can bind to Ca^{2+} /CaM with much higher affinity than the perturbed SOAR.

(11) In a previous report (Qi et al., 2009), 1 μM W7 was enough to inhibit CaM, so what is the justification for using 30/100 μM here? Why not use 1 μM W7? The concentration used here is 30 or 100 times higher than reported. As stated above, W7 is very nonspecific to this high level makes the effect invalid. Also, in Fig. 2F, only 100 μM W7 showed some effect. Again, this effect of W7 does not allow a conclusion. The better control for Fig. 2 is to add 20 mM BAPTA and 100 μM CaM in the pipette solution which is more specific for the inhibition of CaM.

Response:

1) W7 is a canonical CaM inhibitor. The K_d between CaM and W7 is approximately 5-25 μM (Anfinogenova et al, *Cell Physiol. Biochem.*, 11(6): 295-310, 2001) (Dagher et al, *Biochim. Biophys. Acta*, 1793(6): 1068-77, 2009). Different W7 concentrations were used to inhibit endogenous CaM, including 10 μM (Blair et al, *J. Gen. Physiol.*, 133(5): 525-46, 2009) (Sengupta et al, *J. Biol. Chem.*, 282(11): 8474-86., 2007) and 100 μM (Moreau et al, *Curr. Biol.*, 16(16): 1672-7, 2006) (Sullivan et al, *Cell Calcium*, 28(1): 33-46, 2000). Therefore, the concentration of W7 (30/100 μM) in our manuscript is reasonable.

2) As suggested by the reviewer, a slow calcium chelator (10 mM EGTA) and 100 μM CaM were included in the pipette solution for the control experiment. Our results showed that a high concentration of calcium chelator suppressed the CaM-dependent SCDI (see the following Fig.). This figure was also included as **Supplementary Fig. 2** in the revised manuscript.

(12) The FRET data have the same problems as the current data – consistency. In Fig. 5C normalized Eapp was used, while Eapp was used in Fig. 1G and Fig. 6C.

Response: We thank the reviewer for pointing out this problem. In **Fig. 5c**, the normalized Eapp has been replaced with Eapp in the revised manuscript for data consistency.

(13) The ratio of donor/acceptor plays a critical role in determining the final Eapp value. To make an accurate and reliable conclusion, the donor/acceptor ratio for the two compared groups need to be provided and need to be in a similar narrow range. Otherwise, even in the control group, the Eapp will vary a lot. In Fig. 4, the FRET between WT SOAR and WT Orail is about 0.3 Fig. 4B, but it is about 0.24 in Fig. 4G and 4H. This is about a 25% difference. However, in Fig. 4G and 4H, the treatment resulted in about 10% change compared to control and the authors considered this as significant difference. If the variation in the control group is above 20%, it is hard to believe the 10% difference between control group and treated group is significant. Thus, the ratio of donor/acceptor in each individual group is a must. Only cells with similar donor/acceptor ratio are meaningful. The same issue applies for for all other FRET data, such as Fig. 1G, the peak FRET value for CaM-SOAR is about 0.03, which is close to the background FRET value for Orail/SOAR-LF/SS that is about 0.025. Also, the error bar in this figure is huge.

Response:

1) We agree with the reviewer that the donor/acceptor ratio has a great impact on FRET methods. We used the CFP/YFP pair in our experiments, which is commonly used in FRET methods for investigating protein interactions *in vivo*. To make sure that the results are reliable, the cells that were picked for analysis fulfilled the criteria that $0.5 <$

$R_{DA} < 2$, which was recommended in a previous report (Berney et al., *Biophys. J.*, **84(6): 3992-4010, 2003**).

2) The FRET differences between SOAR-YFP and Orai1-CFP among **Figs. 4b, g and h** were due to different extracellular calcium concentrations in the experiments. In **Fig. 4b**, the FRET was recorded without extracellular calcium (0 Ca^{2+} Ringer's buffer). In **Figs. 4g and h**, the FRETs were recorded with 2 mM extracellular calcium (2 Ca^{2+} Ringer's buffer) for the purpose of CaM activation. In the latter case, calcium influx through the Orai1 channel could induce the dissociation of a part of SOAR-Orai1 complex, which is consistent with a previous report (Navarro-Borelly et al, *J. Physiol.* **586(22): 5383-401, 2008**). Therefore, it is reasonable that the latter E_{app} value is lower than the former one. The different extracellular Ca^{2+} levels were included in the figure legend in the revised manuscript.

3) As noted by the reviewer, the peak E_{app} value for SOAR-CaM in **Fig. 1g** is indeed relatively small, which may be due to the large amount of endogenous CaM. The endogenous CaM will compete with overexpressed CaM-CFP to bind SOAR-YFP and thus decrease the FRET signal between CFP and YFP.

(14) On page 11, line 4 from bottom, "After 5 minutes of extracellular Ca^{2+} influx, the FRET E_{app} value at the PM was significantly higher in W-7-treated cells than in DMSO-treated cells (Figure 4H)". It would definitely have been more convincing if the FRET value at time 0 was provided. This is an important criticism.

Response: In its resting state, CaM is not loaded with calcium. Upon SOCE activation, massive calcium flux into the cell occurs through the Orai1 channel. Then, CaM senses the calcium elevation in the cytosol and binds to SOAR. Thus, there will be a response time between calcium influx and CaM regulation. Waiting five minutes after calcium influx gave a relatively stable FRET value.

*Reviewer #3 (Remarks to the Author):
The manuscript (NCOMMS-17-05452-T) presents a wealth of structural and functional data (both in vitro and in vivo) to convincingly demonstrate that Ca^{2+} -bound CaM*

(Ca/CaM) binds to the protein Stim1 and promotes Ca²⁺-induced inactivation of the Orai channel, which is important for Store-operated Ca²⁺ entry (SOCE). This work is potentially significant to the field and should be published. Overall, the functional and structural data and analysis all appear satisfactory. However, I have a few comments and questions below about the NMR structural analysis that should be addressed before a final recommendation can be made.

1. Supplemental Figure 2 shows a very nice HSQC spectrum of ¹⁵N-labeled STIM1 (363-416) construct, and a titration of this spectrum with Ca/CaM reveals key residues of STIM1 (A369, G379 and I383) that likely make contact to Ca/CaM (Fig. 3C). It would be good to also show (in Fig S2) an overlay of the HSQC spectrum of ¹⁵N-labeled STIM1 (363-416) construct in the presence of saturating unlabeled Ca/CaM to better illustrate which residues are affected the most upon Ca/CaM binding. In other words, Fig S2 should show spectrum of STIM1 alone in black on top of a spectrum of STIM1 in presence CaM in red to illustrate which STIM1 residues are most affected by CaM binding.

Response: As suggested by the reviewer, we added the overlay spectra of ¹⁵N-STIM1 in the absence and presence of Ca²⁺/CaM in **Supplementary Fig. 3** in the revised manuscript, and the residues that were most strongly affected are illustrated.

2. In a separate supplemental figure, I recommend showing HSQC spectrum of ¹⁵N-labeled Ca/CaM in the presence of unlabeled STIM1(363-416). This spectrum would then identify the residues in CaM that are most affected by STIM1 binding. The chemical shift perturbation data for both CaM and STIM1 could then be used to guide the docking of the known structures of CaM and STIM1(363-416) solved previously by Yang et al in Proc Natl Acad Sci U S A. 2012 Apr 10;109(15):5657-62. An atomic-level structural model of the CaM/STIM1 complex would be a nice addition to this paper and would not require that much extra work.

Response: As suggested by the reviewer, we have titrated ¹⁵N-CaM/Ca²⁺ with unlabeled STIM1 (363-416) wild type and the L390S/F391S mutant using NMR spectroscopy. The NMR spectra were added as **Supplementary Fig. 4** in the revised manuscript. The residues in CaM that were most affected by STIM1 titration were identified. We also generated a 3D structural model of CaM/Ca²⁺ with STIM1 (**Supplementary Fig. 5**) with the HADDOCK program using the chemical shift perturbations and the published PDB structures of STIM1 (PDB code: 3TEQ) and CaM/Ca²⁺ (PDB code: 1EXR).

3. Fig. 3c and 3d show NMR titration of Ca/CaM binding to the ¹⁵N-STIM1 (363-416)

mutant L390S-F391S. The K_d values determined for the different residues seem to be rather different (138 μM for A369 vs 71 μM for I383). What is the explanation/interpretation for the different K_d values here? Do the different K_d values indicate two different sites? The ITC binding data in Fig. 1B indicate there is only one site. So, it is not clear why two different K_d values are measured for the two residues. There needs to be a more detailed molecular interpretation of the different K_d values here and what is the functional significance?

Response: The K_d values that were determined by the NMR methodology may vary for each residue due to the diverse local conformational changes during the formation of the protein complex. The various K_d values from residues A369, G379 and I383 may be related to their individual locations in the binding interface; the residue A369 is further away from the binding interface, while residues G379 and I383 are much closer to the binding site. The result that residues that are located in different environments lead to different K_d values from NMR analysis may shed light on the intrinsic dynamics of the protein complex that is formed. We averaged the K_d values of the three residues to estimate the binding affinity between these two proteins. Our personal biased view is that there is only one binding site, as shown in the ITC measurement (**Fig. 1b**) and modeling studies (**Supplementary Fig. 5**).

4. For the NMR titration in Figs. 3c and 3d, it would be good to also show the corresponding NMR titration data for wild type STIM1 binding to CaM. Does the NMR titration for wild type show fast exchange (ie single peak that moves like in Fig. 3c) or is it slow exchange (ie two separate peaks) due to the higher binding affinity?

Response: As suggested by the reviewer, we performed new NMR experiments that probed the interactions of ^{15}N -CaM/ Ca^{2+} with STIM1 wild type and the L390S/F391S mutant. The residues that are vicinal to the binding interface show slow-to-intermediate exchange, while the residues that are located further away from the binding interface exhibit the characteristics of fast exchange in the ^{15}N -HSQC spectra. The addition of greater than 1.2 equivalents of STIM1 over ^{15}N -CaM/ Ca^{2+} did not change the ^{15}N -HSQC spectra of ^{15}N -CaM/ Ca^{2+} further (**Supplementary Fig. 4a**), indicating that the binding affinity of CaM/ Ca^{2+} with wild-type STIM1 (363-416) is relatively high. In contrast, the interaction of STIM1 mutant L390S/F391S with CaM appeared to be in

the fast exchange regime in the HSQC spectra, and the addition of L390S/F391S gradually changed the ^{15}N -HSQC spectrum of CaM/Ca $^{2+}$ (**Supplementary Fig. 4b**), suggesting that the binding affinity of CaM/Ca $^{2+}$ with STIM1 mutant L390S/F391S is much weaker.

5. Does the STIM1 SOAR region (residues 363-416) contain a known calmodulin binding motif? If so, what type of CaM motif is it (IQ, 1-12 etc)? There needs to be more discussion about the residues in STIM1 that interact with CaM, and relate the amino acid sequence of CaM binding site in STIM1 with other known CaM binding motifs.

Response: To the best of our knowledge, the SOAR region (residues 363-416) does not contain a known calmodulin-binding motif. We thus searched for the CaM-binding motif within SOAR in a CaM target database (**Yap et al., J. Struct. Funct. Genomics., 1(1): 8-14, 2000**) and found that the region near L390/F391 represents the most likely binding site. We then validated this prediction through IP and FRET experiments and further located the specific binding site using NMR titration. So far, two regions in STIMs were found to interact with Ca $^{2+}$ /CaM (**Bauer et al., Biochemistry, 47(23): 6089-6091, 2008**). One is located in the STIM1 C-terminal region, and the other is located in the STIM2 SOAR domain. First, the C-terminal polybasic region (667-685) in STIM1 was found to interact with Ca $^{2+}$ /CaM by using ITC and NMR methods. This region (667-685) did not contain any known CaM-binding site and consists of several basic and hydrophobic residues. ITC titration of CaM with the polybasic peptide revealed a strong binding affinity (1 μM) (**Bauer et al., Biochemistry, 47(23): 6089-6091, 2008**). Second, the 459-482 region in SOAR2 of STIM2, which corresponds to residues 368-391 in SOAR1 of STIM1, was found, through pull-down and SPR methods, to interact with Ca $^{2+}$ /CaM (**Miederer et al., 6:6899, Nat. Commun., 2014**). This region is also rich in basic and hydrophobic residues and did not contain any known CaM-binding motif. We have included this information in the **Discussion** section of the revised manuscript.

REVIEWERS' COMMENTS:

Reviewer #1 (Remarks to the Author):

The authors have adequately addressed most of my concerns and the manuscript appears strengthened.

There are two small points I still would like to be addressed:

- 1) I do now understand the protocol to elicit Orai1 currents in electrophysiological experiments with SOAR. So experiments depicted in Fig. 2 are now clear. I only found the experiments shown in Suppl Fig. 1 at first sight somewhat misleading, as my first impression was that the current sizes at negative potential do not match those under the various conditions shown in Fig. 2. The reason is that Fig. 2 depicts normalized currents, while Suppl. Fig. 1 shows currents of individual cells. This should be made clear in the legend of Suppl. Fig. 1.
- 2) Regarding the use of overexpressed CaM versus CaM infusion via pipette, as was my original question 2, there are a number examples both with voltage-dependent or TRPV6 Ca²⁺ channels that show that overexpression of YFP-tagged CaM apparently slows its outward diffusion in whole-cell experiments enabling successful recordings over 200s. The author may consider at least a trial.

Reviewer #2 (Remarks to the Author):

This reviewer only commented privately to the editors, stating that despite your great efforts to respond to this reviewer's previous concerns, the reviewer was not satisfied. The main reason for this is that in the reviewer view, the results reported in your work are in potential conflict with previous studies showing that a STIM1 phenylalanine residue that is crucial for Orai1 interaction is located very close to the calmodulin binding site. In the opinion of this reviewer, any mutations in this region will impair the interaction between the sensor and channel.

Reviewer #3 (Remarks to the Author):

The revised manuscript addresses all my previous concerns and should now be suitable for publication.

Reviewers' comments:

Reviewer #1 (*Remarks to the Author*):

The authors have adequately addressed most of my concerns and the manuscript appears strengthened.

There are two small points I still would like to be addressed:

1) I do now understand the protocol to elicit Orail currents in electrophysiological experiments with SOAR. So experiments depicted in Fig. 2 are now clear. I only found the experiments shown in Suppl Fig. 1 at first sight somewhat misleading, as my first impression was that the current sizes at negative potential do not match those under the various conditions shown in Fig. 2. The reason is that Fig. 2 depicts normalized currents, while Suppl. Fig. 1 shows currents of individual cells. This should be made clear in the legend of Suppl. Fig. 1.

Response: As suggested by the reviewer, we modified the legend of Supplementary Figure 1 as follows to avoid potential confusion.

“The I-V curves were generated from the peak current induced by the ramp stimulation protocol, representing the mean original current density size of individual cells. The number in the parenthesis indicates the number of cells that were analyzed.”

2) Regarding the use of overexpressed CaM versus CaM infusion via pipette, as was my original question 2, there are a number examples both with voltage-dependent or TRPV6 Ca²⁺ channels that show that overexpression of YFP-tagged CaM apparently slows its outward diffusion in whole-cell experiments enabling successful recordings over 200s. The author may consider at least a trial.

Response: As suggested by the reviewer, we measured the SCDI current of mCherry-CaM overexpressed cells. The effect of overexpressed CaM on SCDI was not as obvious as that of perfused CaM (see the following **Fig.** versus **Fig. 2**), probably due to wash-out effect of overexpressed CaM after cell membrane is broken.

Reviewer #2 (Remarks to the Author):

This reviewer only commented privately to the editors, stating that despite your great efforts to respond to this reviewer's previous concerns, the reviewer was not satisfied. The main reason for this is that in the reviewer view, the results reported in your work are in potential conflict with previous studies showing that a STIM1 phenylalanine residue that is crucial for Orai1 interaction is located very close to the calmodulin binding site. In the opinion of this reviewer, any mutations in this region will impair the interaction between the sensor and channel.

Response: The important point in our paper is that the binding of Ca²⁺-CaM to SOAR disrupts the STIM1-Orai1 complex. Whether or not this binding event affects the residue F394 conformation originally identified by Dr. Don Gill's lab is beyond the scope of this paper and maybe further studied in the future.

Reviewer #3 (Remarks to the Author):

The revised manuscript addresses all my previous concerns and should now be suitable for publication.

Response: We thank the reviewer for positive comment.